# Learning Interpretable Options by Identifying Reward Diffusion Bottlenecks in Reinforcement Learning

**Yiming Fei** [1 2]  **Lang Qin** [1]  **Rui Yan** [3]  **Huajin Tang** [1 2]

## Abstract

Bottleneck states, which connect distinct regions of the state space, provide a principled and interpretable basis for constructing temporal abstractions in Hierarchical Reinforcement Learning (HRL). However, existing bottleneck identification methods primarily rely on topological analysis of the state-transition graph, limiting their scalability to high-dimensional or continuous domains. To address this challenge, we introduce Value Power Strength (VPS), a value function-based metric inspired by the analogy between the Bellman equation and Kirchhoff's current law, to quantify bottleneck property via the diffusion of reward in Markov Decision Processes (MDPs). VPS is estimated efficiently using value functions learned from random reward signals and captures reward diffusion bottlenecks in both discrete and continuous state spaces. Leveraging VPS, we design options that guide agents toward or away from bottleneck regions. Experimental results on classic tabular domains, continuous-control Point-Maze, and Atari 2600 games demonstrate that the VPS-based framework discovers semantically meaningful subgoals and substantially improves exploration efficiency.

## 1. Introduction

Reinforcement Learning (RL) has achieved remarkable progress, but sparse rewards and long horizons remain challenging (Sutton & Barto, 1998; Ecoffet et al., 2021). Hierarchical Reinforcement Learning (HRL) addresses these issues by decomposing tasks through action abstractions or subgoals, reducing exploration complexity and improving sample efficiency (Pateria et al., 2021; Nachum et al., 2019). Hierarchical agents have shown strong gains across navigation, manipulation, and visual-control benchmarks (Nachum et al., 2018; Park et al., 2023; Eysenbach et al., 2018; Bagaria et al., 2023).

Within HRL, the option framework represents temporally extended behavior as a policy with initiation and termination conditions, enabling high-level planning over multi-step units (Sutton et al., 1999). Option discovery methods either exploit state-space structure, such as graph/topology or spectral embeddings, to select subgoals (Menache et al., 2002; Şimşek et al., 2005; Machado et al., 2017; Klissarov & Machado, 2023), or optimize explicit objectives, such as planning or cover time, with theoretical guarantees (Solway et al., 2014; Jinnai et al., 2019a;b; Ivanov et al., 2025). Structure-driven approaches often yield interpretable options aligned with salient landmarks, while objective-driven approaches emphasize provable efficiency.

Bottleneck states—bridges between weakly coupled regions of the state space—have clear semantics and are natural intrinsic targets for option design. Existing methods usually cast the MDP as a graph and score states with topological criteria such as betweenness centrality or cut-based measures (Freeman, 1977; Newman, 2005; Brandes, 2001; Shi & Malik, 2000). Although effective in tabular settings, they require explicit graph construction and compute scores outside the value-learning loop, limiting scalability and integration with function approximation.

Recent work interprets MDP value functions as discrete potential fields: the Bellman equation relates to Poisson-type equations on the state-transition graph, graph Laplacians, and random walks (Mahadevan & Maggioni, 2007; Machado et al., 2017; Wang et al., 2021). Circuit theory offers a concrete analogue, where Kirchhoff's and Ohm's laws define currents, voltages, and power dissipation on weighted graphs (Doyle & Snell, 1984; Chung, 1997; Chandra et al., 1989; Tetali, 1991). This motivates circuit-theoretic, value-based primitives for option discovery.

To obtain a bottleneck indicator within the standard value-learning loop, we exploit the correspondence between the

[1]College of Computer Science and Technology, Zhejiang University, Hangzhou, China [2]The State Key Lab of Brain-Machine Intelligence, Zhejiang University, Hangzhou, China [3]College of Computer Science and Technology, Zhejiang University of Technology, Hangzhou, China. Correspondence to: Huajin Tang <htang@zju.edu.cn>.

*Proceedings of the 43 rd International Conference on Machine Learning*, Seoul, South Korea. PMLR 306, 2026. Copyright 2026 by the author(s).

Bellman equation and Kirchhoff's current law: in resistive networks, flow conservation and local power dissipation reveal constrained transport. We map node power to a value-based quantity and propose *Value Power Strength* (VPS), which characterizes bottlenecks of reward diffusion without explicit state-transition graphs. VPS is estimated by injecting multiple independent random rewards, learning the induced value functions, and computing VPS scores; the resulting field defines an intrinsic potential for options that steer agents toward or away from high-VPS states. Because it relies only on standard value estimation, VPS extends naturally to high-dimensional settings via function approximation.

Our main contributions are: (1) we propose VPS, a value function–based metric for state-level bottlenecks without explicit graph analysis; (2) we derive estimation procedures for discrete and continuous state spaces and prove almost sure convergence to the true VPS under mild assumptions; and (3) we design a VPS-based option discovery scheme that yields semantically meaningful, interpretable temporally extended actions and improves exploration across tabular, continuous-control, and high-dimensional visual domains.

## 2. Related Work

VPS is related to Laplacian-based representation learning, where value functions can be viewed as potentials on the state-transition graph and their Dirichlet energy characterizes graph structure. Early proto-value functions (PVFs) use Laplacian eigenvectors as basis functions (Mahadevan, 2005; Mahadevan & Maggioni, 2007). Eigenoptions construct options from Laplacian eigenvectors to follow globally smooth directions (Machado et al., 2017), and later work connects eigenoptions to successor representations (Machado et al., 2018; 2023). Recent advances scale Laplacian learning to high-dimensional settings (Wu et al., 2019; Gomez et al., 2024) and study broader spectral operator learning (Touati et al., 2023; Ryu et al., 2025), enabling online spectral option discovery (Klissarov & Machado, 2023) and improved credit assignment (Kotamreddy & Machado, 2025).

Among these spectral option-discovery methods, Eigenoptions are the most direct point of comparison for VPS. A key difference is that eigenoptions are not bottleneck-specific: as the number of eigenvectors (hence options) increases, the extrema of different eigenvectors typically spread across the graph, yielding heterogeneous targets (e.g., room centers, edges, and sometimes passages). Although the Fiedler vector may align with a dominant coarse bottleneck, bottlenecks are generally multi-scale and non-unique, and it is unclear a priori which eigenvectors correspond to bottlenecks at which scales. VPS instead aggregates value-based quantities induced by multiple random rewards into a single

bottleneck score, so even with many options the semantics remain anchored to bottleneck states.

VPS also connects to auxiliary-task methods that learn many value predictions from random cumulants (Lyle et al., 2021; Zheng et al., 2021; Ramesh et al., 2022). Proto-Value Networks scale this idea to many auxiliary heads, where the span of random-cumulant value predictions approaches leading successor/Laplacian eigenspaces (Farebrother et al., 2023). Unlike these approaches that use auxiliary values primarily as features, we use random rewards diagnostically to construct a bottleneck-oriented scalar (VPS) and derive options that explicitly steer toward or away from bottlenecks.

## 3. Preliminaries

RL focuses on training agents to make sequential decisions by interacting with an environment. The process is commonly modeled as an MDP (Puterman, 1994), defined by a tuple $(\mathcal{S}, \mathcal{A}, \mathcal{P}, r, \gamma)$. At each time step $t$, the agent at state $S_t \in \mathcal{S}$ takes an action $A_t \in \mathcal{A}$ and the next state $S_{t+1} \in \mathcal{S}$ is determined by the transition probability kernel $\mathcal{P}(s'|s, a) = \Pr(S_{t+1} = s'|S_t = s, A_t = a)$. Then the environment generates an immediate reward $R_{t+1}$. The objective of the agent is to learn a policy $\pi : \mathcal{S} \times \mathcal{A} \to [0, 1]$ maximizing the expected discounted cumulative rewards $G_t = \mathbb{E}\left[\sum_{k=0}^{\infty} \gamma^k R_{t+k+1}\right]$ where $\gamma \in [0, 1)$ is a discount factor.

### 3.1. Bellman Expectation Equation.

Value function-based RL methods (Mnih et al., 2015; Hessel et al., 2018) aim to estimate the value function, which represents the expected return starting from a given state or state-action pair. The state-value function is defined as $V_\pi(s) = \mathbb{E}_\pi[G_t|S_t = s]$. For a stationary policy $\pi$, the state–value function satisfies

$$V_\pi(s) = \sum_{a \in \mathcal{A}} \pi(a|s) \sum_{s' \in \mathcal{S}} \mathcal{P}(s'|s, a)\big[r(s, a, s') + \gamma V_\pi(s')\big], \tag{1}$$

which is also known as the Bellman expectation equation.

For finite $\mathcal{S} = \{s_1, \dots, s_{|\mathcal{S}|}\}$, let $V_\pi$ and $r_\pi$ collect state values and expected rewards, and define $[P_\pi]_{ij} = \sum_{a \in \mathcal{A}} \pi(a|s_i)\mathcal{P}(s_j|s_i, a)$. Then (1) becomes

$$\mathcal{T}V_\pi = r_\pi, \tag{2}$$

where $\mathcal{T} = I - \gamma P_\pi$ is the linear Bellman equation operator induced by $\pi$. For $\gamma < 1$, the closed-form solution is obtained as

$$V_\pi = (I - \gamma P_\pi)^{-1} r_\pi. \tag{3}$$

## 3.2. Random Rewards via QR Decomposition

For tabular implementation, we generate a set of diverse random rewards by orthogonalizing Gaussian samples. Let $n = |\mathcal{S}|$ and sample a reward matrix $R \in \mathbb{R}^{n \times k}$ with i.i.d. entries $R_{si} \sim \mathcal{N}(0, \sigma^2)$. We then apply QR decomposition

$$R = Q\, \widetilde{R}, \qquad Q^\top Q = I_k, \qquad (4)$$

and use the $i$-th column of $Q$ as the $i$-th random reward $r^{(i)}(s) = Q_{si}$. This QR-based construction preserves the rotational invariance of Gaussian sampling (hence no preferred direction in expectation) while guaranteeing within-batch orthogonality and reducing redundant, highly correlated rewards (Mezzadri, 2006).

## 3.3. Random Rewards via Random Fourier Features

In high-dimensional or continuous spaces, we construct random rewards using Random Fourier Features (RFF) (Rahimi & Recht, 2007) on the learned representation $z_\phi(s)$. Let $k(z, z') = \exp\big(-\|z - z'\|^2 / (2\ell^2)\big)$ be the RBF kernel with length-scale $\ell > 0$. We sample

$$\mathbf{w}_j \sim \mathcal{N}(0, \ell^{-2} I_d), \qquad b_j \sim \mathrm{Uniform}(0, 2\pi),$$

and define the $K$-dimensional randomized feature map

$$\zeta_j(s) = \sqrt{\tfrac{2}{K}} \, \cos\big(\mathbf{w}_j^\top z_\phi(s) + b_j\big), \quad j = 1, \dots, K.$$

The $m$-th random reward is a random linear combination of these features,

$$r^{(m)}(s) = \sum_{j=1}^{K} a_j^{(m)} \zeta_j(s), \qquad a_j^{(m)} \sim \mathcal{N}(0, 1) \ \text{ i.i.d.}$$
$$(5)$$

As $K \to \infty$, $r^{(m)}$ converges (in mean-square) to a sample from a zero-mean Gaussian process $\mathcal{GP}(0, k)$ with the kernel $k$ specified above.

## 3.4. Laplacian of State Transition Graphs.

The state transition graph of an MDP $(\mathcal{S}, \mathcal{A}, \mathcal{P}, r, \gamma)$ provides a structural representation of the environment by modeling the transitions between states as a graph (Şimşek & Barto, 2008). Formally, a state transition graph $\mathcal{G} = (V, E, W)$ is a weighted directed graph where the vertices $V = \mathcal{S}$ correspond to the set of states and the directed edges $E \subseteq \mathcal{S} \times \mathcal{S}$ represent corresponding state transitions. An edge $(s, s') \in E$ exists if there exists an action $a \in \mathcal{A}$ such that the transition probability $\mathcal{P}(s'|s, a) > 0$. The weight function $W : E \to \mathbb{R}^+$ assigns each edge a positive value that captures the possibility of the corresponding transition.

For an MDP with a finite state set, we embed its transition dynamics in a weighted graph whose adjacency matrix is

$W = [W_{ss'}]_{s,s' \in \mathcal{S}}$ and whose diagonal degree matrix is

$$D = \mathrm{diag}\big(d(s_1), \dots, d(s_{|\mathcal{S}|})\big), \quad d(s) = \sum_{s'} W_{ss'}. \quad (6)$$

For the Laplacian construction, $W$ denotes an undirected weighted graph, or the symmetrized version of the directed transition graph above. The topological structure of the graph is then characterized by the random-walk Laplacian

$$L_{\mathrm{rw}} = I - D^{-1} W, \qquad (7)$$

whose eigenvectors associated with the smallest (non-zero) eigenvalues minimize the discrete Dirichlet energy and therefore vary smoothly within highly connected regions while changing sharply across narrow bridges (Von Luxburg, 2007; Belkin & Niyogi, 2003).

## 3.5. Option Framework.

The option framework, introduced by (Sutton et al., 1999), extends RL by enabling agents to perform temporally extended actions, facilitating hierarchical decision-making. An option is defined as a tuple $\mathcal{O} = (\mathcal{I}_o, \pi_o, \beta_o)$, where $\mathcal{I}_o \subseteq \mathcal{S}$ is the initiation set specifying where the option can start, $\pi_o : \mathcal{S} \times \mathcal{A} \to [0, 1]$ is the intra-option policy governing behavior during the option's execution, and $\beta_o : \mathcal{S} \to [0, 1]$ is the termination condition defining the probability of ending the option at each state. Agents execute options by selecting an option, following its policy, and terminating with $\beta_o$.

# 4. Option Discovery via Value Power Strength

Unlike traditional graph-based methods that compute bottleneck metrics with transition graphs, our approach leverages the structural analogy between the Bellman equation and Kirchhoff's current law to propose *Value Power Strength* (VPS)—a value function–based bottleneck metric. VPS enables the construction of intrinsic rewards that guide option learning toward or away from bottleneck regions.

## 4.1. Value Power Strength of States

Consider the single-source single-sink resistance network illustrated in Figure 1. Current flow through a node is a natural bottleneck score (Newman, 2005):

$$I_i = \frac{1}{2} \sum_j W_{ij} |U_i - U_j|, \qquad i \notin \{s, g\}. \quad (8)$$

where $W_{ij} = 1/R_{ij}$ is the edge conductance, $U_i$ is the electric potential, and $1/2$ avoids double-counting undirected edges. The source and sink currents are defined separately as $I_s = I_g = I_{in}$, so node current measures the share of the overall charge-transport task borne by each vertex. Bottlenecks can also be viewed through energy conversion: if

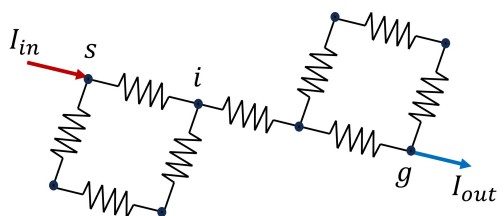

*Figure 1.* Single-source single-sink resistance network

the network is regarded as a heater, nodes with larger power dissipation bear a larger portion of the transport-induced energy-conversion load. We therefore use node power as a uniform, differentiable bottleneck quantity:

$$P_i = \frac{1}{2} \sum_j W_{ij} (U_i - U_j)^2. \tag{9}$$

Node power locally decomposes the graph Dirichlet energy (Chung, 1997):

$$\mathcal{E}(U) = \frac{1}{2} \sum_{i,j} W_{ij} (U_i - U_j)^2 = \frac{1}{2} U^\top L_c U = \sum_i P_i, \tag{10}$$

where $L_c$ is the combinatorial Laplacian. Large $P_i$ therefore marks where energy concentrates.

To avoid explicit state-transition graphs, we map node power to RL through the Bellman–Kirchhoff analogy. Kirchhoff's law is

$$L_c U = I, \tag{11}$$

where $U$ and $I$ are node-potential and external-current vectors, and $L_c = D - W$ is the combinatorial Laplacian. Together with the Bellman linear system (2), this is a discrete analogue of the Poisson equation

$$\nabla^2 \phi = -\rho, \tag{12}$$

where $\phi$ is a potential and $\rho$ is a source. Thus, electric potential $U(s)$ corresponds to value $V(s)$, and external current corresponds to reward; Table 1 summarizes this correspondence.

Mapping node power through this analogy yields the following value-based bottleneck metric. The coefficient $1/2$ is omitted since it is shared by all states.

**Definition 1** (Value Power Strength). Consider an MDP $(\mathcal{S}, \mathcal{A}, \mathcal{P}, r, \gamma)$ with a stationary policy $\pi$. The *Value Power Strength* (VPS) of state $s \in \mathcal{S}$ is defined as the expected squared difference in value between $s$ and its successor states:

$$\varphi_\pi(s) = \mathbb{E}_{s' \sim P_\pi(\cdot|s)} \left[ (V_\pi(s) - V_\pi(s'))^2 \right]. \tag{13}$$

When the state space $\mathcal{S}$ is finite, VPS can be formulated as

$$\varphi_\pi(s) = \sum_{s' \in \mathcal{S}} P_\pi(s' \mid s) (V_\pi(s) - V_\pi(s'))^2 \tag{14}$$

where $P_\pi(s' \mid s)$ is the policy-induced transition probability, given for discrete action spaces by $P_\pi(s' \mid s) = \sum_{a \in \mathcal{A}} \pi(a \mid s) \mathcal{P}(s' \mid s, a)$.

Analogous to node power in (9), VPS measures how strongly a state contributes to reward propagation as the value function is formed by the Bellman equation. High-VPS states therefore indicate bottlenecks of reward diffusion in the state space.

### 4.2. Online Estimate of VPS

Since the value of VPS depends on the value function $V_\pi$, we have proposed a simultaneous update law of $V_\pi$ and $\varphi_\pi$ as below.

$$\begin{cases} V_{t+1}(S_t) \leftarrow V_t(S_t) + \alpha_t [R_{t+1} + \gamma V_t(S_{t+1}) - V_t(S_t)] \\ \varphi_{t+1}(S_t) \leftarrow \varphi_t(S_t) + \beta_t \left[ (V_t(S_t) - V_t(S_{t+1}))^2 - \varphi_t(S_t) \right] \end{cases} \tag{15}$$

The convergence of the update law is proved by Proposition 1 in Appendix A.1.

Although (15) offers an online algorithm for VPS estimation, its definition still depends on a specific policy $\pi$ and reward $r$. To preliminarily validate the effectiveness of VPS for bottleneck identification, we consider a MiniGrid environment "Rooms" which contains different types of bottlenecks (Figure 2). The agent starts from random positions, each episode has a fixed length, and follows a random walk policy with four primitive actions (up, down, left and right). For rewards, we examine: (i) assigning a reward of 1 to a few designated states (see "Value Power Strength" in Figure 2); and (ii) sequentially assigning a reward of 1 to each grid and aggregating the resulting VPS distributions (see "Cumulative Value Power Strength"). As shown in Figure 2, placing a reward near a bottleneck effectively highlights it via VPS, while the cumulative VPS closely matches the patterns of current flow and shortest path betweenness centrality (Brandes & Fleischer, 2005; Borgatti & Everett, 2006).

### 4.3. VPS-Based Option Discovery

Based on the above, VPS can reveal reward diffusion bottlenecks in MDPs under random walk policies with suitable reward design. However, two challenges remain for option discovery: (i) how to design rewards to estimate VPS; (ii) how to design options with VPS.

For the first challenge, we propose an orthogonal random reward construction based on QR decomposition. For each batch, we first sample a Gaussian reward matrix $R \in \mathbb{R}^{|\mathcal{S}| \times k}$ with i.i.d. entries and then orthogonalize its columns:

$$R_{si} \sim \mathcal{N}(0, \sigma^2) \text{ i.i.d.}, \qquad R = Q\widetilde{R}, \qquad r^{(i)}(s) = Q_{si}. \tag{16}$$

Here $Q^\top Q = I$ so that the resulting rewards are mutually

*Table 1.* Analogy between Kirchhoff's current law, Bellman equation, and Poisson's equation.

| No. | Kirchhoff's Current Law | Bellman Equation | Poisson's Equation |
|-----|-------------------------|------------------|--------------------|
| 1 | External current input $I$ | External reward input $r_\pi$ | External source term $-\rho$ |
| 2 | Node voltage $U$ | State value $V_\pi$ | Scalar potential $\phi$ |
| 3 | Graph Laplacian operator $L_c$ | Linear Bellman equation operator $\mathcal{T}$ | Laplacian operator $\nabla^2$ |
| 4 | Conductance matrix $W$ | State-transition matrix $P_\pi$ | — |

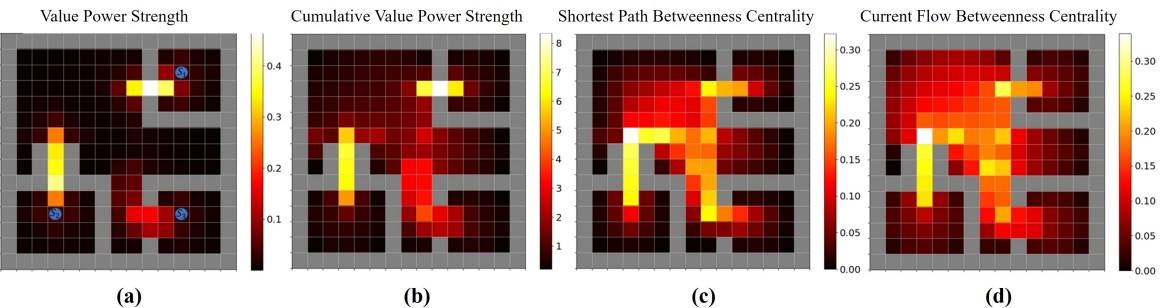

| (a) | (b) | (c) | (d) |
|-----|-----|-----|-----|

*Figure 2.* **VPS and betweenness centrality distribution in MiniGrid-Rooms**: (a) VPS distribution when the agent performs a random walk and only three selected grids provide a visit reward of 1 (all other grids give reward 0). (b) Cumulative VPS map obtained by sequentially assigning a visit reward of 1 to each grid (one target grid per experiment, all others 0), computing VPS for each case, and aggregating the resulting VPS fields. (c) Shortest-path betweenness centrality. (d) Current-flow betweenness centrality.

orthogonal within the batch. The rationality of this design can be understood from the following spectral result.

**Lemma 1.** *Let $(\mathcal{S}, \mathcal{A}, \mathcal{P}, r, \gamma)$ be a finite MDP with stationary policy $\pi$ such that the induced Markov chain is reversible. Let $P_\pi \in \mathbb{R}^{|\mathcal{S}| \times |\mathcal{S}|}$ denote the reversible policy-induced transition matrix, which is self-adjoint under the stationary-distribution weighted inner product, and define the random walk Laplacian as $L_{\mathrm{rw}} = I - P_\pi$. The linear Bellman equation operator is denoted by $\mathcal{T} = I - \gamma P_\pi$ with $0 < \gamma < 1$. Then, for every eigenpair $(\mu_k, v_k)$ of $L_{\mathrm{rw}}$, such that $L_{\mathrm{rw}} v_k = \mu_k v_k$, the vector $v_k$ is also an eigenvector of $\mathcal{T}$, with corresponding eigenvalue $\lambda_k = 1 - \gamma(1 - \mu_k)$.*

*Proof.* See Appendix A.2. □

**Theorem 1** (Spectral Solution of the State-Value Function). *Consider a finite, reversible MDP as in Lemma 1, and let $P_\pi$ be the reversible policy-induced transition matrix with unique stationary distribution $\mathbf{d} \in \mathbb{R}^{|\mathcal{S}|}$ (i.e., $P_\pi^\top \mathbf{d} = \mathbf{d}$, $\sum_i d_i = 1$, $d_i > 0$). Under reversibility, $P_\pi$ is self-adjoint under the $\mathbf{d}$-weighted inner product. Define the random walk Laplacian as $L_{\mathrm{rw}} = I - D^{-1}W = I - P_\pi$, where $D = \mathrm{diag}(\mathbf{d})$ and $W = DP_\pi$. Let $\{(\mu_k, v_k)\}_{k=1}^{|\mathcal{S}|}$ be the eigenpairs of $L_{\mathrm{rw}}$ with $\{v_k\}$ forming an orthonormal basis under the $\mathbf{d}$-weighted inner product:*

$$\langle f, g \rangle_{\mathbf{d}} := \sum_{i=1}^{|\mathcal{S}|} d_i f(s_i) g(s_i).$$

*Let $\mathbf{r} \in \mathbb{R}^{|\mathcal{S}|}$ denote the reward vector whose $i$-th entry $\mathbf{r}_i$ is the expected immediate reward at $s_i$. Then, the solution*

*$V$ to the Bellman equation $\mathcal{T}V = \mathbf{r}$ admits the following spectral decomposition:*

$$V = \sum_{k=1}^{|\mathcal{S}|} \frac{1}{1 - \gamma(1 - \mu_k)} \langle \mathbf{r}, v_k \rangle_{\mathbf{d}} \, v_k. \qquad (17)$$

*Proof.* See Appendix A.3. □

The use of QR-orthogonalized random rewards for VPS is motivated by the spectral result of Theorem 1. In (16), we first sample a Gaussian reward matrix and then apply QR to obtain an orthonormal set of reward vectors (columns of $Q$). This construction inherits the rotational invariance of Gaussian sampling (hence it does not prefer any particular eigenvector direction in expectation), while the orthogonality constraint reduces redundancy among rewards within a batch (Mezzadri, 2006). Consequently, the state-value function $V$ is dominated by low-frequency components—those with small $\mu_k$—due to the spectral scaling $[1 - \gamma(1 - \mu_k)]^{-1}$, while high-frequency eigenvectors are suppressed. This ensures $V$ reflects the large-scale structure of the state space, resulting in a smooth partitioning (Chung, 1997).

The reversibility assumption is used only for this clean spectral interpretation of QR rewards; it is not required by the definition of VPS, the online estimator, or the option-learning pipeline. More generally, for any finite policy-induced chain, let $G = (I - \gamma P_\pi)^{-1}$ and let $g(s)$ be the row of $G$ corresponding to state $s$. For isotropic QR reward probes $q_i$, where $\mathbb{E}_Q$ denotes expectation over the random QR reward matrix $Q = [q_1, \ldots, q_k]$, the expected averaged

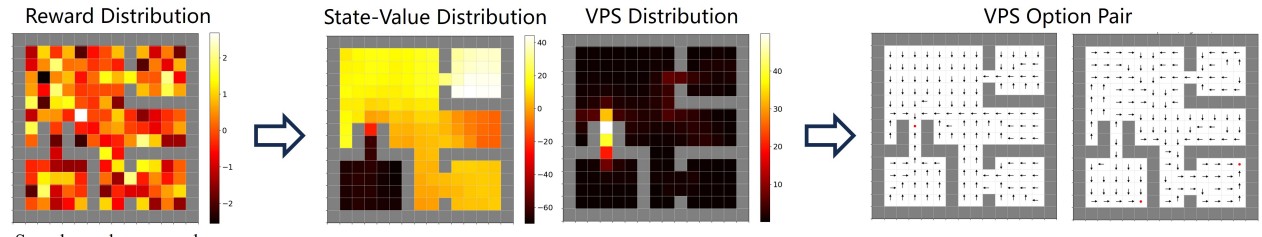

*Figure 3.* Pipeline for VPS-based option discovery

VPS satisfies

$$\mathbb{E}_Q\left[\frac{1}{k}\sum_i \varphi_{q_i}(s)\right] = \frac{1}{|\mathcal{S}|}\sum_{s'} P_\pi(s'|s)\|g(s)-g(s')\|_2^2. \tag{18}$$

Thus, in non-reversible MDPs, QR rewards still probe sharp one-step changes in discounted future reachability. High-VPS states therefore include classical undirected bottlenecks as well as directed transition interfaces, such as one-way transitions or irreversible object-state changes.

For the second challenge, inspired by the Eigenoption (Machado et al., 2018), the VPS-based dual potential-difference intrinsic rewards $r_{\text{int}}^{(m,\xi)}(s,s') = \xi\left(\varphi^{(m)}(s') - \varphi^{(m)}(s)\right)$ with $\xi \in \{+1,-1\}$ are considered to obtain options towards or away from bottlenecks, where the superscript $(m)$ denotes the $m$-th pair of options corresponding to the $m$-th random reward sampling.

The $m$-th pair of options is constructed by maximizing the expected discounted cumulative intrinsic reward $r_{\text{int}}^{(m,\xi)}(s,s')$ under the option discount $\gamma_Q$, with the intra-option policy $\pi_o$ and termination rule $\beta_o$ parameterized for each option $o$. The optimal option-value function is defined as

$$Q_o^*(s,a) = \max_{\pi_o} \mathbb{E}\left[\sum_{t=0}^{\infty} \gamma_Q^t r_{\text{int}}^{(m,\xi)}(S_t, S_{t+1}) \Big| S_0 = s, A_0 = a, o\right]. \tag{19}$$

The corresponding greedy intra-option policy is given by $a_o^*(s) = \arg\max_{a\in\mathcal{A}} Q_o^*(s,a)$.

The initiation set of each option is set as the entire state space $\mathcal{S}$. The termination probability $\beta_o(s)$ is defined as

$$\beta_o(s) = \begin{cases} 1, & \text{if } Q_o^*(s,a) < 0, \quad for \quad \forall\, a \in \mathcal{A} \\ 1/N, & \text{otherwise} \end{cases} \tag{20}$$

where $N$ is a manually specified positive constant. The options designed in this manner enable the agent to reach states with the highest or lowest VPS values (or local extrema, depending on the discount factor).

Figure 3 illustrates the overall procedure of VPS-based option discovery, while the algorithmic procedure is provided in Appendix B.

### 4.4. Function Approximation Cases

To generalize VPS to high-dimensional or continuous state spaces, we learn a shared encoder $z_\phi(s) \in \mathbb{R}^d$ with a value head $V_\theta$ and a VPS head $\varphi_\psi$. We train $V_\theta$ with a target encoder $z_{\bar{\phi}}$ and target value head $V_{\bar{\theta}}$ and estimate VPS as a conditional regression target built from $V_{\bar{\theta}}$:

$$\mathcal{L}_V(\theta,\phi) = \mathbb{E}_{(s,r,s')\sim\mathcal{D}}\left[r + \gamma V_{\bar{\theta}}(z_{\bar{\phi}}(s')) - V_\theta(z_\phi(s))\right]^2, \tag{21}$$

$$\mathcal{L}_\varphi(\psi,\phi) = \mathbb{E}_{(s,s')\sim\mathcal{D}}\left[\left(V_{\bar{\theta}}(z_{\bar{\phi}}(s)) - V_{\bar{\theta}}(z_{\bar{\phi}}(s'))\right)^2 - \varphi_\psi(z_\phi(s))\right]^2. \tag{22}$$

In practice, we optionally stop the gradient from $\mathcal{L}_\varphi$ to $z_\phi$ if training is unstable.

For high-dimensional or continuous spaces, we use Random Fourier Features (RFF) (Rahimi & Recht, 2007) to construct random rewards on the representation $z_\phi(s)$ (see Preliminaries and (5)). Building rewards directly on $z_\phi(s)$ makes the induced random fields continuous in $z_\phi$. When representation learning aligns $z_\phi$ with transition dynamics, this continuity serves as a practical proxy for smoothness on the state-transition graph, which stabilizes VPS estimation. A promising future direction is to design random rewards that match transition geometry more faithfully (e.g., successor-feature or diffusion kernels on $z_\phi$).

## 5. Experimental Results

We conduct experiments on tabular MiniGrid-KeyLock and Taxi-v3, continuous-control PointMaze-FourRooms, and Atari 2600 environments. In all experiments, agents collect transitions under a random walk policy, and VPS is trained with random rewards (16) and (5).

### 5.1. Tabular Cases

For the tabular cases, we evaluate on MiniGrid-KeyLock (Chevalier-Boisvert et al., 2023) and Gym Taxi-v3 (Brockman et al., 2016). We compare four settings: **Primitive** (no options; only primitive actions), **Random Option**

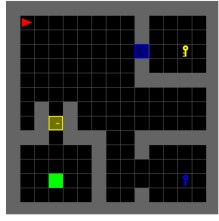 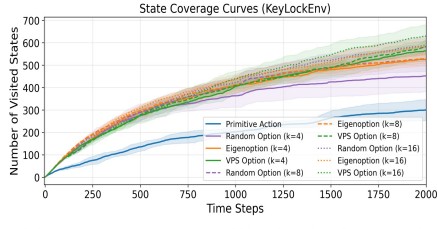 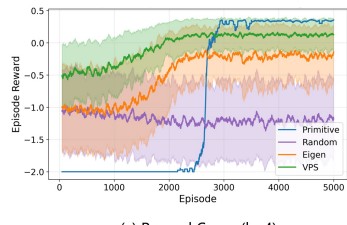

(a)MiniGrid-KeyLock     (b) State coverage curves for different exploration methods     (c) Reward Curve (k=4)

*Figure 4.* (a) MiniGrid-KeyLock environment. (b) State coverage under uniform selection over primitive actions and startable options, averaged over 5 seeds. (c) Q-learning with option exploration ($k$=4 options).

(assign each state a Gaussian random potential and train an option pair to reach the max/min-potential states), **Eigenoption** (implemented following Machado et al. (2017)), and **VPS Option** (our method).

**MiniGrid-KeyLock.** To evaluate whether VPS options improve exploration, we first consider a gridworld with a blue/yellow key, matching doors, and a goal; the agent starts at $(1, 1)$. Observations provide global object-state values (agent position/orientation, door open/closed, key on-map vs. carried/consumed). The action space has 6 actions (up/down/left/right, pick up, toggle door). Reward is $+1$ at the goal (terminate), otherwise $-0.01$ per step. Detailed experimental settings are provided in Appendix C.1.

Figure 5 visualizes phase-conditioned VPS heatmaps for a representative option in MiniGrid-KeyLock. High-VPS regions concentrate around doors, keys, and key-dependent transition states, showing that VPS captures task-semantic bottlenecks beyond geometric passages.

We report two main MiniGrid-KeyLock results: (i) state coverage under option-based random walk (Figure 4(b)); and (ii) Q-learning with option exploration (Figure 4(c)). VPS consistently improves coverage and learning speed. Additional goal-reaching success distributions under random walk are reported in Appendix C.2, with the detailed settings in Appendix C.1. The **Primitive** return may become higher later since we plot training returns (no separate evaluation), and executing options can increase steps-to-goal.

**Taxi-v3**. To evaluate whether VPS options can participate in temporal credit assignment, we use Taxi-v3 as a semi-Markov decision process (SMDP) benchmark. During reward collection, the agent follows an $\epsilon$-greedy policy over an augmented action set: exploration samples uniformly from primitive actions and learned options, and selected options are assigned credit through SMDP Q-learning. Each option set comprises 20 options derived from 10 intrinsic rewards; we compare against 20 eigenoptions (from the 10 smallest Laplacian eigenvalues) and 20 random options. All option discovery methods are repeated with 10 different random seeds, resulting in 10 independent sets of options, and each set is evaluated over 10 training runs. Interestingly,

a considerable proportion of VPS options are able to complete the pick-up and drop-off task without ever receiving external rewards during training, as shown in Figure 6(a). This phenomenon arises because bottleneck states—where the taxi, passenger, and destination coincide—are automatically identified. Figures 6(b) and 6(c) report the frequency of automatic task completion and reward acquisition via Q-learning. Overall, VPS options effectively capture meaningful behaviors and facilitate reward acquisition, although adding options accelerates learning but does not always improve final performance.

### 5.2. Continuous-Control PointMaze

To evaluate whether VPS options remain useful beyond discrete state-action spaces, we consider a continuous-control PointMaze-FourRooms task, where narrow passages form natural bottlenecks. During reward-free pretraining, VPS first collects 500,000 random-walk transitions to fit RFF value/VPS estimators and then trains eight SAC option policies, while DIAYN trains eight reward-free skills. Figure 7 shows that RFF-induced value functions are smooth over the maze, whereas VPS concentrates near passage-like bottlenecks and yields positive/negative options with complementary rollouts.

For downstream evaluation, we compare SAC (Haarnoja et al., 2018), SAC with DIAYN-assisted exploration (Eysenbach et al., 2018), and SAC with VPS-assisted exploration. In the sequential waypoint task (Figure 8(a)), episodes start from $(2, 2)$, last at most 1000 steps, and give reward $+1$ only when four waypoints are visited in order. Detailed downstream and intervention settings are provided in Appendix C.1, Table 11. In this experiment, VPS options and DIAYN skills are used only as exploration tools and do not participate in credit assignment; the RL process remains a standard MDP rather than an SMDP, and SAC updates still optimize the external waypoint reward. Figure 8(b) shows that VPS options achieve higher mean evaluation reward, indicating that bottleneck-oriented skills provide useful temporal abstraction. Representative DIAYN rollouts are shown in Appendix Figure 17.

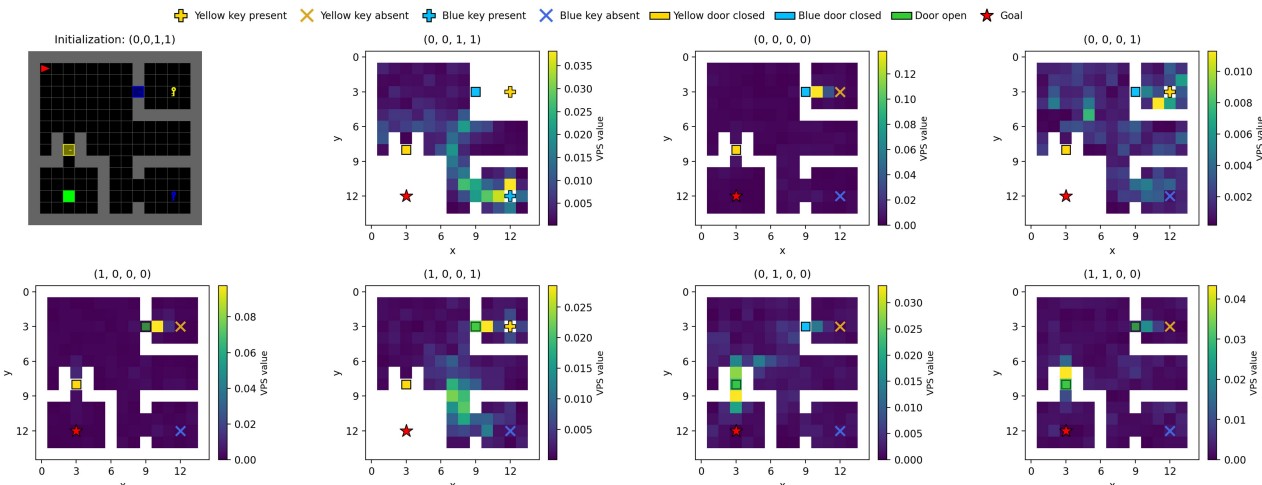

*Figure 5.* VPS heatmaps for a representative option in MiniGrid-KeyLock. The top-left panel shows the initialization rendering. The remaining seven panels are phase-conditioned heatmaps titled by their $(a, b, c, d)$ configuration, where $(a, b, c, d) = $ (blue_door_open, yellow_door_open, blue_key_on_map, yellow_key_on_map). Each heatmap value at $(x, y)$ is obtained by averaging the values of all reachable $(x, y, \text{direction})$ states and projecting that mean onto position $(x, y)$; brighter regions indicate stronger reward-diffusion bottlenecks.

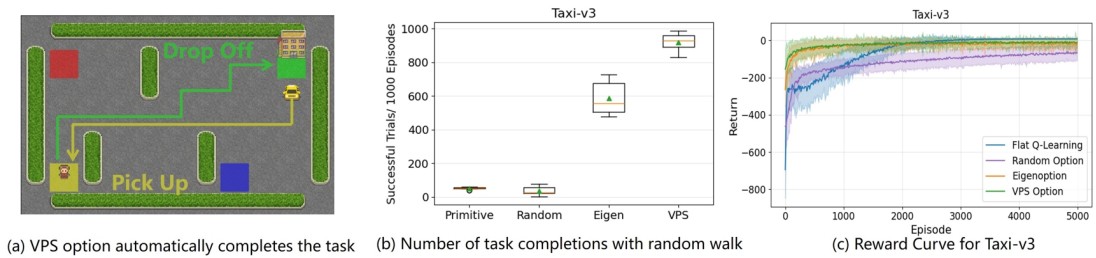

(a) VPS option automatically completes the task    (b) Number of task completions with random walk    (c) Reward Curve for Taxi-v3

*Figure 6.* Performance of different options in Taxi-v3. A "successful trial" refers to an episode in which the taxi successfully transports a passenger from the starting point to the destination once.

## 5.3. Atari 2600

To verify whether VPS can identify bottlenecks in high-dimensional visual observations, we consider Atari 2600 games. The agent performs random walks to collect transition data, and we train 8 different VPS estimators with RFF rewards using single-frame $84 \times 84$ grayscale images as input. We then evaluate human-gameplay videos in which a player guides the agent to explore the state space. For each frame, we compute its $\phi$ value using the trained VPS networks. Figure 9 illustrates that in the ALE/Venture-v5 environment, VPS peaks reliably correspond to interpretable bottleneck frames—specifically, doorways between different rooms (the positions of the agent are marked by white cross stars). Additional results for more Atari games are provided in Appendix C.2.

## 6. Limitations and Future Work

A practical limitation of VPS is computational cost: estimating bottlenecks from many random reward functions may require many corresponding value predictions, and a direct implementation scales linearly with the number of reward probes. A natural remedy is to share computation across probes by learning a common transition-aware encoder and temporal-difference targets, while using lightweight reward-specific heads to predict each induced value field and VPS target. This preserves reward diversity while avoiding repeated learning of the same transition structure.

Successor representations provide a principled route for such sharing, since they separate environment dynamics from rewards and allow values to be recomputed when reward weights change (Dayan, 1993; Barreto et al., 2017; 2018). Future work could combine VPS with successor-feature modules, deep successor representation learning (Kulkarni et al., 2016), or Proto-Value Networks (Farebrother et al., 2023) to compute many reward-conditioned value functions and VPS maps in parallel. Such shared representations may also improve stability by regularizing noisy value estimates, which otherwise can create spurious high-VPS regions.

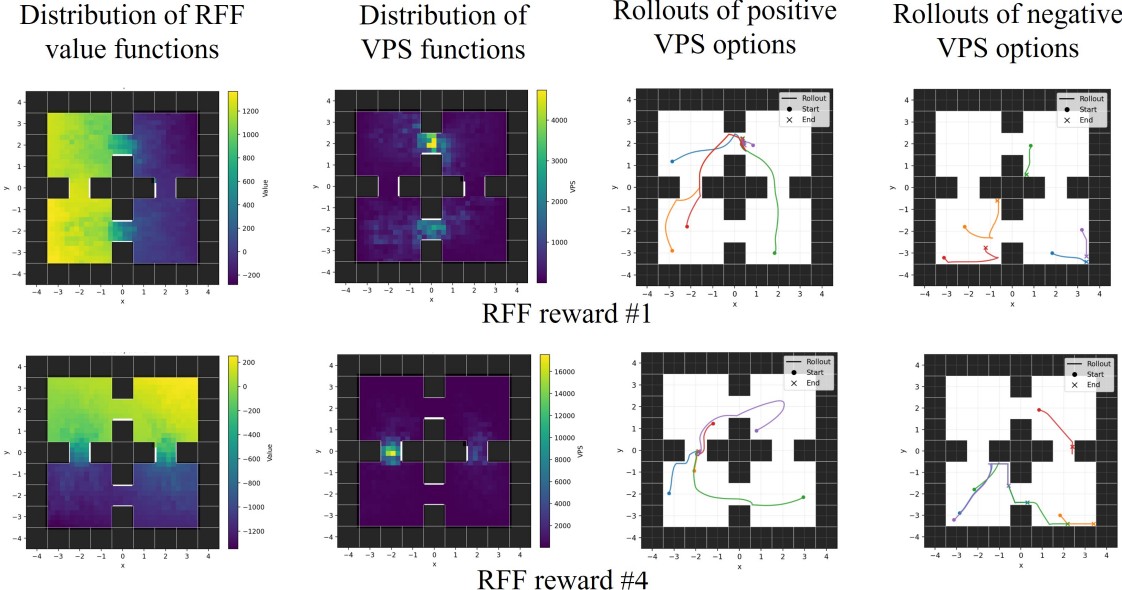

*Figure 7.* VPS estimation and option rollouts in continuous-control PointMaze-FourRooms. For two RFF reward instances, the learned value functions diffuse smoothly through the maze, while VPS concentrates near narrow passages that mediate reward propagation. Positive and negative VPS options induce complementary rollouts toward different extrema of the VPS field.

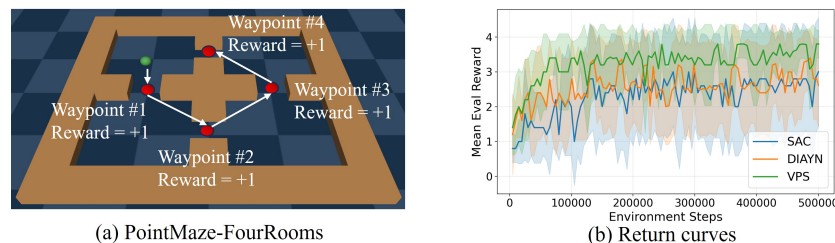

*Figure 8.* Downstream evaluation in PointMaze-FourRooms. (a) Environment and sequential waypoint task: the agent starts from $(2, 2)$ and receives reward $+1$ only when visiting the four waypoints in order. (b) Mean evaluation reward over downstream environment steps for SAC, SAC with DIAYN-assisted exploration, and SAC with VPS-assisted exploration. Pretrained DIAYN skills and VPS options are invoked only as exploration interventions during SAC training.

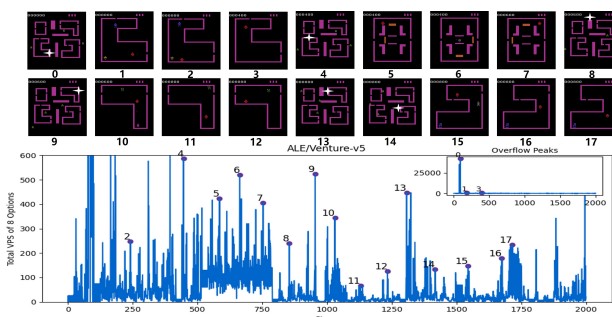

*Figure 9.* Bottlenecks identified through VPS in ALE/Venture-v5

## 7. Conclusion

This paper introduces Value Power Strength (VPS), a value function–based metric for identifying reward-diffusion bot-tlenecks without constructing an explicit transition graph. Motivated by the analogy between node-wise power dissipation and squared value discrepancies, VPS provides a local signal that can be estimated in tabular domains and with function approximation. We further use VPS as a potential to learn interpretable option pairs that steer toward or away from high-VPS regions.

The resulting framework links bottleneck discovery, value estimation, and option learning within a unified pipeline that can extend beyond explicit graph representations. Empirically, VPS highlights task-relevant transition regions such as doors, key-dependent states, narrow passages, and salient Atari frames, suggesting that reward-diffusion structure is a useful principle for interpretable temporal abstraction in sparse-reward domains.

## Acknowledgments

This work was supported in part by the National Natural Science Foundation of China under Grant Nos. 62236007, 32441113, and 62276235. The authors sincerely thank the anonymous reviewers and chairs for their constructive comments, which helped improve the quality of this work.

## Impact Statement

This work aims to improve interpretable and sample-efficient hierarchical reinforcement learning by identifying reward-diffusion bottlenecks. Its potential risks are those common to reinforcement learning methods: better exploration and temporal abstraction should be deployed with task-specific safety constraints in sensitive or real-world settings.

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

# Appendix Contents

# A. Proofs of Main Results

## A.1. Proposition 1 and its proof

$$
\begin{cases}
V_{t+1}(S_t) \leftarrow V_t(S_t) + \alpha_t \left[ R_{t+1} + \gamma V_t(S_{t+1}) - V_t(S_t) \right] \\
\varphi_{t+1}(S_t) \leftarrow \varphi_t(S_t) + \beta_t \left[ (V_t(S_t) - V_t(S_{t+1}))^2 - \varphi_t(S_t) \right]
\end{cases}
\tag{23}
$$

**Proposition 1.** *Consider a finite MDP defined by the tuple $(\mathcal{S}, \mathcal{A}, \mathcal{P}, r, \gamma)$ with a stationary policy $\pi$. Let $\{V_t\}_{t \geq 0}$ and $\{\varphi_t\}_{t \geq 0}$ be the sequences of value estimates and VPS estimates updated according to (23). Suppose the following conditions hold:*

1. *The Markov chain induced by $\pi$ is ergodic (irreducible and aperiodic);*

2. *The learning rates $\{\alpha_t\}$ and $\{\beta_t\}$ satisfy the Robbins-Monro conditions:*

$$
\sum_t \alpha_t = \sum_t \beta_t = \infty, \quad \sum_t \alpha_t^2 < \infty, \quad \sum_t \beta_t^2 < \infty;
$$

3. *The second moment of the reward is uniformly bounded:*

$$
\mathbb{E}_\pi \left[ R_{t+1}^2 \mid S_t = s \right] \leq R_{\max}^2 < \infty, \quad \forall s \in \mathcal{S}.
$$

*Then, for $\forall s \in \mathcal{S}$, the sequences $\{V_t\}$ and $\{\varphi_t\}$ converge almost surely to the true state-value function $V_\pi$ and the true VPS function $\varphi_\pi$, respectively.*

*Proof.* The convergence of the value function sequence $\{V_t\}$ to $V_\pi$ under the given conditions is a well-established result (Tsitsiklis & Van Roy, 1996). We thus focus on proving the almost sure convergence of the VPS sequence $\{\varphi_t\}$ to $\varphi_\pi$.

### Step 1: Preliminaries and Notation

Since the Markov chain induced by $\pi$ is ergodic, every state $s \in \mathcal{S}$ is visited infinitely often almost surely. Let $\{\tau_k(s)\}_{k=1}^\infty$ denote the increasing sequence of time steps at which state $s$ is visited. Consider the subsequence $\{\varphi_{\tau_k}(s)\}_{k \geq 1}$.

It is standard that, under the Robbins-Monro conditions and the assumptions on rewards, $V_t(s) \to V_\pi(s)$ almost surely for all $s \in \mathcal{S}$. Since $\mathcal{S}$ is finite, this convergence is uniform:

$$
\forall \varepsilon > 0, \ \exists K(\varepsilon) \text{ such that } \forall k \geq K(\varepsilon), \ \max_{s \in \mathcal{S}} |V_{\tau_k}(s) - V_\pi(s)| < \varepsilon \quad \text{a.s.}
\tag{24}
$$

### Step 2: Stochastic Update for VPS

By definition, when $s$ is visited at time $\tau_k$, the update is:

$$
\varphi_{\tau_k+1}(s) = \varphi_{\tau_k}(s) + \beta_{\tau_k} \left[ (V_{\tau_k}(s) - V_{\tau_k}(S_{\tau_k+1}))^2 - \varphi_{\tau_k}(s) \right]
$$
$$
= (1 - \beta_{\tau_k}) \varphi_{\tau_k}(s) + \beta_{\tau_k} (V_{\tau_k}(s) - V_{\tau_k}(S_{\tau_k+1}))^2.
$$

Let $e_k(s) := \varphi_{\tau_k}(s) - \varphi_\pi(s)$ denote the error at the $k$-th visit to $s$. We have

$$
e_{k+1}(s) = (1 - \beta_{\tau_k}) e_k(s) + \beta_{\tau_k} \left[ (V_{\tau_k}(s) - V_{\tau_k}(S_{\tau_k+1}))^2 - \varphi_\pi(s) \right]
$$
$$
= (1 - \beta_{\tau_k}) e_k(s) + \beta_{\tau_k} \left( A_k + B_k \right)
\tag{25}
$$

where we decompose

$$
A_k := (V_{\tau_k}(s) - V_{\tau_k}(S_{\tau_k+1}))^2 - (V_\pi(s) - V_\pi(S_{\tau_k+1}))^2,
$$
$$
B_k := (V_\pi(s) - V_\pi(S_{\tau_k+1}))^2 - \mathbb{E}_{s' \sim P_\pi(\cdot|s)} \left[ (V_\pi(s) - V_\pi(s'))^2 \right].
$$

**Step 3: Uniform Control of the Bias Term** $A_k$

Let $\delta_1 = V_{\tau_k}(s) - V_\pi(s)$, $\delta_2 = V_{\tau_k}(S_{\tau_k+1}) - V_\pi(S_{\tau_k+1})$. Then,

$$
\begin{aligned}
A_k &= (V_{\tau_k}(s) - V_{\tau_k}(S_{\tau_k+1}))^2 - (V_\pi(s) - V_\pi(S_{\tau_k+1}))^2 \\
&= \left((V_\pi(s) + \delta_1) - (V_\pi(S_{\tau_k+1}) + \delta_2)\right)^2 - (V_\pi(s) - V_\pi(S_{\tau_k+1}))^2 \\
&= \left((V_\pi(s) - V_\pi(S_{\tau_k+1})) + (\delta_1 - \delta_2)\right)^2 - (V_\pi(s) - V_\pi(S_{\tau_k+1}))^2 \\
&= 2(V_\pi(s) - V_\pi(S_{\tau_k+1}))(\delta_1 - \delta_2) + (\delta_1 - \delta_2)^2.
\end{aligned}
$$

Applying triangle inequality and uniform convergence (24), when $k$ is large enough,

$$
|\delta_1|, |\delta_2| < \varepsilon, \quad |V_\pi(s) - V_\pi(S_{\tau_k+1})| \le 2M,
$$

where $M = \max_{s \in \mathcal{S}} |V_\pi(s)|$. Thus,

$$
|A_k| \le 2 \cdot 2M \cdot 2\varepsilon + (2\varepsilon)^2 = 8M\varepsilon + 4\varepsilon^2.
$$

Therefore, $A_k \to 0$ almost surely as $k \to \infty$.

**Step 4: Martingale Difference Property of** $B_k$

For each $k$, $B_k$ is a martingale difference: conditioned on $\mathcal{F}_{\tau_k}$ (the $\sigma$-algebra up to time $\tau_k$),

$$
\mathbb{E}[B_k \mid \mathcal{F}_{\tau_k}] = 0.
$$

Moreover, since $|V_\pi(s)| \le M$, it follows $|B_k| \le (2M)^2 = 4M^2$ almost surely, so the sequence has uniformly bounded second moment.

**Step 5: Convergence via Auxiliary Sequence Construction**

The error dynamics in (25) can be expressed as:

$$
e_{k+1}(s) = (1 - \beta_{\tau_k})e_k(s) + \beta_{\tau_k} w_k + \beta_{\tau_k} \delta_k \tag{26}
$$

where:

$$
\begin{aligned}
w_k &:= B_k + (A_k - \mathbb{E}[A_k \mid \mathcal{F}_{\tau_k}]) \\
\delta_k &:= \mathbb{E}[A_k \mid \mathcal{F}_{\tau_k}]
\end{aligned}
$$

By construction, $w_k$ satisfies:

(a) $\mathbb{E}[w_k \mid \mathcal{F}_{\tau_k}] = 0$  (martingale difference)

(b) $|w_k| \le |B_k| + |A_k| + \mathbb{E}[|A_k| \mid \mathcal{F}_{\tau_k}] \le 4M^2 + 16M\varepsilon + 8\varepsilon^2 < \infty$ a.s.

Thus $\mathbb{E}[w_k^2 \mid \mathcal{F}_{\tau_k}] \le C$ for some $C < \infty$ almost surely.

Consider the auxiliary sequence defined by:

$$
\widetilde{e}_{k+1}(s) = (1 - \beta_{\tau_k})\widetilde{e}_k(s) + \beta_{\tau_k} w_k, \quad \widetilde{e}_0(s) = e_0(s) \tag{27}
$$

To apply Lemma 1 in (Tsitsiklis & Van Roy, 1996), the following conditions should be satisfied:

(a) $\mathbb{E}[w_k \mid \mathcal{F}_{\tau_k}] = 0$

(b) $\mathbb{E}[w_k^2 \mid \mathcal{F}_{\tau_k}] \le C < \infty$

(c) $\beta_{\tau_k} \in [0, 1]$ (by assumption)

(d) $\sum_{k=0}^{\infty} \beta_{\tau_k} = \infty$ (Robbins-Monro)

(e) $\sum_{k=0}^{\infty} \beta_{\tau_k}^2 < \infty$ (Robbins-Monro)

While Lemma 1 requires $\beta_{\tau_k} \in [0,1]$, our sequence may have $\beta_{\tau_k} > 1$ for some $k$. However, since $\sum \beta_{\tau_k}^2 < \infty$, we have $\beta_{\tau_k} \to 0$ a.s. Thus for any $\epsilon > 0$, there exists $K_\epsilon$ such that for $k \geq K_\epsilon$, $\beta_{\tau_k} < \epsilon$. We can choose $\epsilon < 1$ so that for $k \geq K_\epsilon$, $\beta_{\tau_k} \in [0,1]$.

For $k < K_\epsilon$, the finite number of updates where $\beta_{\tau_k} \geq 1$ do not affect almost sure convergence. Specifically, we can restart the sequence at $k = K_\epsilon$ with initial condition $e_{K_\epsilon}(s)$, which is bounded a.s. by the finite state space assumption. Therefore, without loss of generality, we assume $\beta_{\tau_k} \in [0,1]$ for all $k$.

Under these conditions, Lemma 1 implies $\widetilde{e}_k(s) \to 0$ almost surely as $k \to \infty$.

The difference between the sequences satisfies:

$$|e_k(s) - \widetilde{e}_k(s)| \leq \sum_{m=0}^{k-1} \beta_{\tau_m} |\delta_m| \prod_{j=m+1}^{k-1} (1 - \beta_{\tau_j}).$$

Fix any $\eta > 0$. Since $\delta_m \to 0$ almost surely (from Step 3), there exists $K$ such that $|\delta_m| \leq \eta$ for all $m \geq K$ on the almost sure event of convergence. Splitting the sum at $K$ gives

$$|e_k(s) - \widetilde{e}_k(s)| \leq \sum_{m=0}^{K-1} \beta_{\tau_m} |\delta_m| \prod_{j=m+1}^{k-1} (1 - \beta_{\tau_j})$$
$$+ \eta \sum_{m=K}^{k-1} \beta_{\tau_m} \prod_{j=m+1}^{k-1} (1 - \beta_{\tau_j}).$$

For each fixed $m < K$, the product term converges to zero because $\sum_j \beta_{\tau_j} = \infty$ and, after the finite restart above, $\beta_{\tau_j} \in [0,1]$. Hence the finite early sum vanishes as $k \to \infty$. The tail sum is bounded by

$$\sum_{m=K}^{k-1} \beta_{\tau_m} \prod_{j=m+1}^{k-1} (1 - \beta_{\tau_j}) = 1 - \prod_{j=K}^{k-1} (1 - \beta_{\tau_j}) \leq 1,$$

so $\limsup_{k \to \infty} |e_k(s) - \widetilde{e}_k(s)| \leq \eta$. Since $\eta$ is arbitrary, $|e_k(s) - \widetilde{e}_k(s)| \to 0$ almost surely.

Combining with $\widetilde{e}_k(s) \to 0$ a.s., we conclude $e_k(s) \to 0$ almost surely.

**Step 6: Extension to the Full Sequence**

The above argument holds for each $s \in \mathcal{S}$. Since the state space is finite and $\varphi_t(s)$ is only updated at visits to $s$, the entire sequence $\{\varphi_t(s)\}_{t \geq 0}$ converges almost surely to $\varphi_\pi(s)$. This completes the proof. $\square$

## A.2. Proof of Lemma 1

**Lemma 1.** *Let $(\mathcal{S}, \mathcal{A}, \mathcal{P}, r, \gamma)$ be a finite MDP with stationary policy $\pi$ such that the induced Markov chain is reversible. Let $P_\pi \in \mathbb{R}^{|\mathcal{S}| \times |\mathcal{S}|}$ denote the reversible policy-induced transition matrix, which is self-adjoint under the stationary-distribution weighted inner product, and define the random walk Laplacian as $L_{\mathrm{rw}} = I - P_\pi$. The linear Bellman equation operator is denoted by $\mathcal{T} = I - \gamma P_\pi$ with $0 < \gamma < 1$. Then, for every eigenpair $(\mu_k, v_k)$ of $L_{\mathrm{rw}}$, such that $L_{\mathrm{rw}} v_k = \mu_k v_k$, the vector $v_k$ is also an eigenvector of $\mathcal{T}$, with corresponding eigenvalue $\lambda_k = 1 - \gamma(1 - \mu_k)$.*

*Proof.* Let $(\mu_k, v_k)$ be an eigenpair of the random walk Laplacian $L_{\mathrm{rw}}$, satisfying:

$$L_{\mathrm{rw}} v_k = \mu_k v_k$$

By definition $L_{\mathrm{rw}} = I - P_\pi$, we have:

$$(I - P_\pi) v_k = \mu_k v_k \tag{28}$$

Rearranging terms yields:

$$P_\pi v_k = (1 - \mu_k)v_k \tag{29}$$

Now consider the linear Bellman equation operator $\mathcal{T} = I - \gamma P_\pi$:

$$\mathcal{T}v_k = (I - \gamma P_\pi)v_k = v_k - \gamma P_\pi v_k$$

Substituting (29):

$$\mathcal{T}v_k = v_k - \gamma(1 - \mu_k)v_k = [1 - \gamma(1 - \mu_k)]\, v_k$$

Thus, $v_k$ is an eigenvector of $\mathcal{T}$ with eigenvalue $\lambda_k = 1 - \gamma(1 - \mu_k)$. $\qquad\square$

## A.3. Proof of Theorem 1

**Theorem 1** (Spectral Solution of the State-Value Function). *Consider a finite, reversible MDP as in Lemma 1, and let $P_\pi$ be the policy-induced transition matrix with unique stationary distribution $\mathbf{d} \in \mathbb{R}^{|\mathcal{S}|}$ (i.e., $P_\pi^\top \mathbf{d} = \mathbf{d}$, $\sum_i d_i = 1$, $d_i > 0$). Define the random walk Laplacian as $L_{\mathrm{rw}} = I - D^{-1}W = I - P_\pi$, where $D = \mathrm{diag}(\mathbf{d})$ and $W = DP_\pi$. Let $\{(\mu_k, v_k)\}_{k=1}^{|\mathcal{S}|}$ be the eigenpairs of $L_{\mathrm{rw}}$ with $\{v_k\}$ forming an orthonormal basis under the $\mathbf{d}$-weighted inner product:*

$$\langle f, g \rangle_{\mathbf{d}} := \sum_{i=1}^{|\mathcal{S}|} d_i\, f(s_i)\, g(s_i).$$

*Let $\mathbf{r} \in \mathbb{R}^{|\mathcal{S}|}$ denote the reward vector whose $i$-th entry $\mathbf{r}_i$ is the expected immediate reward at $s_i$. Then, the solution $V$ to the Bellman equation $\mathcal{T}V = \mathbf{r}$ admits the following spectral decomposition:*

$$V = \sum_{k=1}^{|\mathcal{S}|} \frac{1}{1 - \gamma(1 - \mu_k)}\, \langle \mathbf{r}, v_k \rangle_{\mathbf{d}}\, v_k. \tag{30}$$

*Proof.* The Bellman equation for the state-value function is given by:

$$\mathcal{T}V = \mathbf{r}$$

where $\mathcal{T} = I - \gamma P_\pi$ is the linear Bellman equation operator. This can be rewritten as:

$$(I - \gamma P_\pi)V = \mathbf{r} \tag{31}$$

By Lemma 1, for each eigenpair $(\mu_k, v_k)$ of $L_{\mathrm{rw}} = I - P_\pi$, $v_k$ is also an eigenvector of $\mathcal{T}$ with eigenvalue:

$$\lambda_k = 1 - \gamma(1 - \mu_k)$$

Since $P_\pi$ is reversible, it is self-adjoint under the $\mathbf{d}$-weighted inner product. Therefore, $L_{\mathrm{rw}} = I - P_\pi$ and $\mathcal{T} = I - \gamma P_\pi$ are also self-adjoint under this inner product:

$$\langle L_{\mathrm{rw}}u, v \rangle_{\mathbf{d}} = \langle u, L_{\mathrm{rw}}v \rangle_{\mathbf{d}}, \quad \langle \mathcal{T}u, v \rangle_{\mathbf{d}} = \langle u, \mathcal{T}v \rangle_{\mathbf{d}}.$$

Thus $\{v_k\}$ forms an orthonormal basis for $\mathbb{R}^{|\mathcal{S}|}$ under $\langle \cdot, \cdot \rangle_{\mathbf{d}}$:

$$\langle v_i, v_j \rangle_{\mathbf{d}} = \delta_{ij}$$

Expand $V$ and $\mathbf{r}$ in this basis:

$$V = \sum_{k=1}^{|\mathcal{S}|} a_k v_k \tag{32}$$

$$\mathbf{r} = \sum_{k=1}^{|\mathcal{S}|} b_k v_k \tag{33}$$

where coefficients are given by:

$$a_k = \langle V, v_k \rangle_{\mathbf{d}}, \quad b_k = \langle \mathbf{r}, v_k \rangle_{\mathbf{d}}$$

Substitute into (31):

$$(I - \gamma P_\pi) \left( \sum_{k=1}^{|\mathcal{S}|} a_k v_k \right) = \sum_{k=1}^{|\mathcal{S}|} b_k v_k$$

Using the eigenvalue relation from Lemma 1:

$$\sum_{k=1}^{|\mathcal{S}|} a_k \lambda_k v_k = \sum_{k=1}^{|\mathcal{S}|} b_k v_k$$

By orthogonality of $\{v_k\}$, we equate coefficients:

$$a_k \lambda_k = b_k \quad \text{for each } k$$

Thus:

$$a_k = \frac{b_k}{\lambda_k} = \frac{\langle \mathbf{r}, v_k \rangle_{\mathbf{d}}}{1 - \gamma(1 - \mu_k)}$$

Reconstructing $V$:

$$V = \sum_{k=1}^{|\mathcal{S}|} \frac{\langle \mathbf{r}, v_k \rangle_{\mathbf{d}}}{1 - \gamma(1 - \mu_k)} v_k$$

$\square$

## B. Implementation of VPS-Based Option Discovery

---

**Algorithm 1** Online VPS Option Discovery

---

**Require:** Number of random rewards $k$, number of options $2k$, buffer size $B$, Gaussian reward variance $\sigma^2$, VPS learning rates $\alpha$, $\beta$, option Q-learning rate $\eta$, discounts $\gamma_V, \gamma_Q$, termination constant $N$

1: **Initialize** replay buffer $\mathcal{D} \leftarrow \emptyset$
2: **while** $|\mathcal{D}| < B$ **do**
3:     Collect transitions $(s, a, s')$ under a random policy and store them in $\mathcal{D}$
4: **end while**
5: **Stage 1: VPS Estimation**
6: Sample one Gaussian reward matrix $R \in \mathbb{R}^{|\mathcal{S}| \times k}$ and apply QR to obtain orthogonal rewards $\{r^{(i)}\}_{i=1}^{k}$ (columns of $Q$), cf. (16)
7: **for** each $i = 1$ to $k$ **do**
8:     Initialize $V^{(i)}(s)$ and $\varphi^{(i)}(s)$ arbitrarily for all $s$
9:     **repeat**
10:         **for** each sampled transition $(s, a, s') \sim \mathcal{D}$ **do**
11:             $V^{(i)}(s) \leftarrow V^{(i)}(s) + \alpha \big[ r^{(i)}(s) + \gamma_V V^{(i)}(s') - V^{(i)}(s) \big]$
12:             $\varphi^{(i)}(s) \leftarrow \varphi^{(i)}(s) + \beta \big[ (V^{(i)}(s) - V^{(i)}(s'))^2 - \varphi^{(i)}(s) \big]$
13:         **end for**
14:     **until** $V^{(i)}$ and $\varphi^{(i)}$ converge for all $s$
15: **end for**
16: **Stage 2: Option Q-Learning (dual directions)**
17: **for** each $i = 1$ to $k$ **do**
18:     **for** each sign $\xi \in \{+1, -1\}$ **do**
19:         Define intrinsic reward $r_{\text{int}}^{(i,\xi)}(s, s') = \xi \left( \varphi^{(i)}(s') - \varphi^{(i)}(s) \right)$
20:         Initialize $Q_{i,\xi}(s, a)$ arbitrarily for all $s, a$
21:         **repeat**
22:             **for** each sampled transition $(s, a, s') \sim \mathcal{D}$ **do**
23:                $Q_{i,\xi}(s, a) \leftarrow Q_{i,\xi}(s, a) + \eta \Big[ r_{\text{int}}^{(i,\xi)}(s, s') + \gamma_Q \max_{a'} Q_{i,\xi}(s', a') - Q_{i,\xi}(s, a) \Big]$
24:             **end for**
25:         **until** $Q_{i,\xi}$ converges
26:         Define intra-option policy $\pi_{i,\xi}(s) = \arg\max_a Q_{i,\xi}(s, a)$
27:         Define termination $\beta_{i,\xi}(s) = \begin{cases} 1, & \text{if } Q_{i,\xi}(s, a) < 0, \forall a \\ 1/N, & \text{otherwise} \end{cases}$
28:     **end for**
29: **end for**
30: **Return:** $2k$ VPS options $\mathcal{O} = \{(\mathcal{S}, \pi_{i,\xi}, \beta_{i,\xi}) \mid i = 1, \ldots, k; \, \xi = \pm 1\}$

---

For continuous or high-dimensional settings, the same flow is used with RFF rewards from (5) and neural value/VPS losses from Section 4.4 replacing the tabular QR rewards and tabular updates.

## C. Experimental Details and Additional Results

### C.1. Detailed Experimental Settings

All tabular experiments (MiniGrid-KeyLock, Taxi-v3) in this paper were conducted on an Intel i7-13700HX CPU, while all high-dimensional or continuous experiments (PointMaze-FourRooms, Atari 2600 games) were performed on a single Nvidia RTX 4090 GPU. The detailed experimental settings for each environment are as follows.

**MiniGrid-KeyLock**

- Environmental Settings

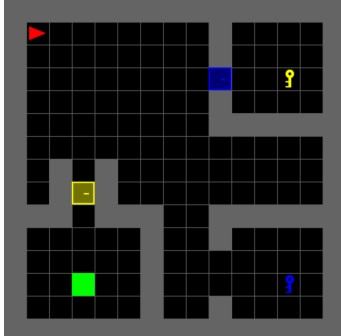

*Figure 10.* MiniGrid-KeyLock

Observation Space: a tuple of discrete variables (`x`, `y`, `direction`, `yellow_door_open`, `blue_door_open`, `yellow_key_on_map`, `blue_key_on_map`), where `x` and `y` are the agent position, `direction` has 4 possible values, each door state is binary (open/closed), and each key state is binary (on the map vs. carried/consumed).

Action Space: 6 discrete actions (up, down, left, right, pick up key, toggle door)

- Option Training phase

| Parameter | VPS Option | Eigenoption | Random Option |
|---|---|---|---|
| Intrinsic reward number ($k_{\text{base}}$) | 4/8/16 | 4/8/16 | 4/8/16 |
| Option number | 8/16/32 | 8/16/32 | 8/16/32 |
| Discount for state-value function ($\gamma_V$) | 0.999 | – | – |
| Discount for Q-learning ($\gamma_Q$) | 0.999 | 0.999 | 0.999 |
| Learning rate for state-value function ($\alpha_V$) | 0.05 | – | – |
| Learning rate for Q-learning ($\alpha_Q$) | 0.1 | 0.1 | 0.1 |
| Buffer episodes ($N_{\text{ep}}$) | 2000 | 2000 | 2000 |
| Episode length ($T_{\text{ep}}$) | 500 | 500 | 500 |

*Table 2.* Hyperparameter settings for option training in MiniGrid-KeyLock.

- Exploration Experiments

MiniGrid-KeyLock exploration includes two experiments. (i) **State coverage experiment** (Figure 4(b)) evaluates how efficiently different option sets cover the state space under option-based random walk. (ii) **Random-walk success-rate test** (Figure 14) measures the probability of reaching the goal under option-based random walk. The experimental configurations for these two exploration experiments are summarized in Table 3 and Table 4, respectively.

| Parameter | Value |
|---|---|
| Number of option sets ($n_{\text{outer}}$) | 5 |
| Runs per option set (inner) | 1 |
| Steps per run ($\max\_\text{steps}$) | 2000 |
| Expected option length $L$ | 15 |

*Table 3.* Core parameters for the state coverage experiment in MiniGrid-KeyLock (Figure 4(b)).

| Parameter | Value |
|---|---|
| Number of option sets ($n_{\text{outer}}$) | 5 |
| Episodes per trial | 10 |
| Maximum steps of an episode ($\max\_\text{steps}$) | 200 |
| Expected option length $L$ | 15 |

*Table 4.* Core parameters for the random-walk-success-rate experiment in MiniGrid-KeyLock (Figure 14).

- Reward Collection Experiments

| Parameter | Value |
|---|---|
| Episodes per run | 5000 |
| Steps per episode | 200 |
| Epsilon-greedy ($\epsilon$) | Annealed from 1 to 0.1 |
| Q-learning learning rate ($\alpha$) | 0.5 |
| Discount factor ($\gamma$) | 0.99 |
| Option horizon $L$ | 15 |
| Runs per option set ($n_{\text{inner}}$) | 1 |
| Number of option sets ($n_{\text{outer}}$) | 5 |
| Reward for reaching goal | 1.0 |
| Step penalty | $-0.01$ |

*Table 5.* Core parameters for the reward collection experiment in MiniGrid-KeyLock.

**Taxi-v3**

- Environmental Settings

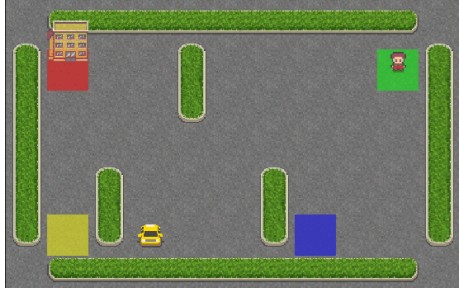

*Figure 11.* Taxi-v3 Environment

Observation Space: 500 discrete states (Taxi position, passenger location, destination)

Action Space: 6 discrete actions (south, north, east, west, pickup, dropoff)

Reward for successful dropoff: 20

Step penalty: $-1$

Invalid pickup/dropoff penalty: $-10$

- Option Training Phase

| Parameter | VPS Option | Eigenoption | Random Option |
|---|---|---|---|
| Intrinsic reward number ($k_{\text{base}}$) | 10 | 10 | 10 |
| Option number | 20 | 20 | 20 |
| Discount for state-value function ($\gamma_V$) | 0.999 | – | – |
| Discount for Q-learning ($\gamma_Q$) | 0.999 | 0.999 | 0.999 |
| Eligibility trace for TD($\lambda$) of $V$ ($\lambda$) | 0.9 | – | – |
| Learning rate for all Q-learning ($\alpha$) | 0.05 | 0.05 | 0.05 |
| Buffer episodes ($N_{\text{ep}}$) | 1000 | 1000 | 1000 |
| Episode length ($T_{\text{ep}}$) | 200 | 200 | 200 |
| Q-learning update steps ($N_{\text{step}}$) | 1,000,000 | 1,000,000 | 1,000,000 |

*Table 6.* Hyperparameter settings for option training in Taxi-v3.

- Random-Walk Task Completion Experiment

| Parameter | Value |
|---|---|
| Number of option sets ($n_{\text{outer}}$) | 10 |
| Episodes per set | 1000 |
| Max steps per episode | 200 |
| Option horizon ($L$) | 10 |
| Success criterion | Pickup and dropoff completed within an episode |
| Action selection | Uniform over primitive actions and options |

*Table 7.* Core parameters for the random walk task completion experiment in Taxi-v3.

- Reward Collection Experiments

In this experiment, Taxi-v3 is treated as an SMDP: the agent follows an $\epsilon$-greedy policy over an augmented action set containing both primitive actions and learned options. During exploration, the exploratory action is sampled uniformly from this augmented action set. When an option is selected, it is executed for at most $L = 10$ primitive steps, and the option-level update uses the cumulative discounted reward collected during execution.

| Parameter | Value |
|---|---|
| Episodes per run | 5000 |
| Steps per episode | 200 |
| Evaluation interval (episodes) | 10 |
| Evaluation trials per interval | 10 |
| $\epsilon$-greedy ($\epsilon$) | 0.1 |
| Q-learning learning rate ($\alpha$) | 0.1 |
| Discount factor ($\gamma$) | 0.95 |
| Option horizon ($L$) | 10 |
| Runs per option set ($n_{\text{inner}}$) | 10 |
| Number of option sets ($n_{\text{outer}}$) | 10 |

*Table 8.* Core parameters for the reward collection experiment in Taxi-v3.

**PointMaze-FourRooms**

- Environmental Settings

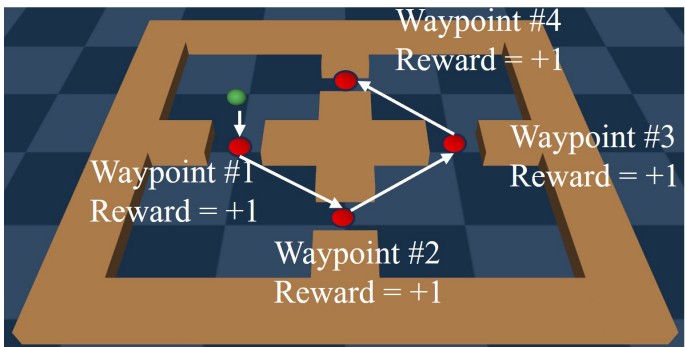

*Figure 12.* PointMaze-FourRooms environment. The point-mass agent navigates through four rooms connected by narrow passages and receives a sequential waypoint reward. Each waypoint gives reward $+1$ when it is reached and then activates the next waypoint.

Environment: continuous-control PointMaze-FourRooms with four rooms connected by narrow passages. These passages form geometric bottlenecks, while the sequential waypoint task requires the agent to visit task-relevant locations in order.

Observation Space: continuous point-mass state features containing position and velocity information.

Action Space: two-dimensional continuous velocity-control actions.

Reward: four sequential waypoints are placed in different rooms. Reaching the currently active waypoint gives reward $+1$ and activates the next waypoint. The downstream episode starts from cell $(2, 2)$, has a maximum horizon of 1000 primitive steps, and terminates successfully after all four waypoints are reached.

- VPS Option Pretraining Phase The main VPS option pretraining parameters are summarized in Table 9.

| Parameter | Value |
|---|---|
| Random-walk transitions | 500,000 |
| Episode length | 300 |
| Number of RFF rewards | 4 |
| RFF frequency scale ($\sigma$) | 0.001 |
| Value discount ($\gamma_V$) | 0.999 |
| Value/VPS learning rate | $1 \times 10^{-3}$ |
| Value/VPS epochs | 50 / 50 |
| Value/VPS batch size | 256 |
| Number of VPS options | $4 \times 2$ |
| Option replay buffer size | 200,000 |
| Option discount ($\gamma_Q$) | 0.99 |
| Option learning rate | $3 \times 10^{-4}$ |

*Table 9.* Core VPS option pretraining parameters for PointMaze-FourRooms.

- DIAYN Skill Pretraining Phase The main DIAYN skill pretraining parameters are summarized in Table 10.

| Parameter | Value |
|---|---|
| Number of skills | 8 |
| Training steps | 500,000 |
| Episode length | 300 |
| SAC discount ($\gamma$) | 0.99 |
| SAC learning rate | $3 \times 10^{-4}$ |
| Replay buffer size | 200,000 |
| Batch size | 256 |
| Discriminator learning rate | $3 \times 10^{-4}$ |
| Discriminator batch size | 256 |

*Table 10.* Core DIAYN skill pretraining parameters for PointMaze-FourRooms.

- Downstream Waypoint Task and Option Intervention The main downstream task and option-intervention parameters are summarized in Table 11.

| Parameter | Value |
| --- | --- |
| Waypoint order | $(4, 2) \rightarrow (6, 4) \rightarrow (4, 6) \rightarrow (2, 4)$ |
| Start cell | $(2, 2)$ |
| Episode length | 1000 |
| Reward per waypoint | $+1$ |
| Maximum return | 4 |
| Training steps | 500,000 |
| Discount factor ($\gamma$) | 0.95 |
| Learning rate | $1 \times 10^{-3}$ |
| Replay buffer size | 500,000 |
| Batch size | 256 |

*Table 11.* Downstream PointMaze-FourRooms task and SAC intervention parameters.

All downstream intervention variants use the pretrained skills/options only to guide exploration; the SAC objective remains the sparse external waypoint reward.

## Atari 2600 Games

- Environmental Settings

  Observation Space: single-frame grayscale images of size $84 \times 84$.

- Network Structure

  The neural networks employed for estimating the value function, VPS, and the intra-policy of options in Atari 2600 games share an identical CNN feature extraction backbone, as illustrated in Figure 13(a). The task-specific heads for each network are depicted in Figure 13(b,c). Here, $S_t$ denotes a preprocessed, single-frame grayscale image of size $84 \times 84$.

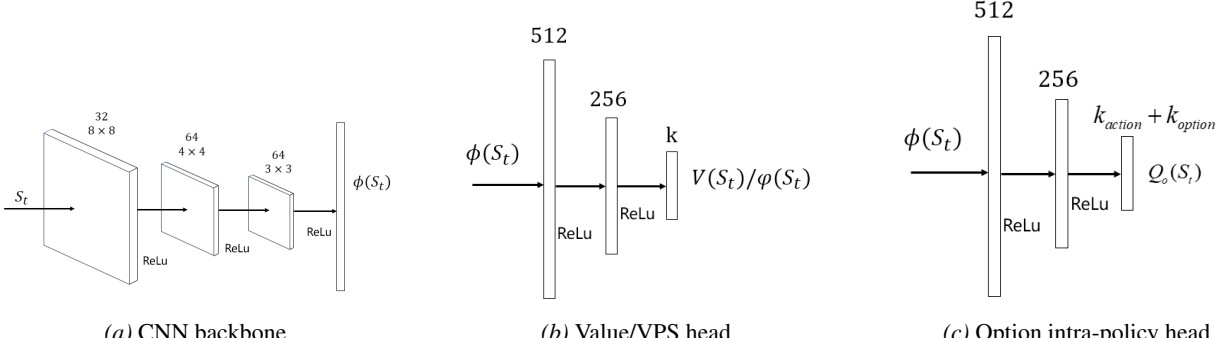

*(a)* CNN backbone  *(b)* Value/VPS head  *(c)* Option intra-policy head

*Figure 13.* Network architectures for Atari 2600 games.

- Hyperparameter Settings

| Parameter | Value |
|---|---|
| Number of options ($k_{\mathrm{opt}}$) | 8 |
| Discount for VPS ($\gamma_V$) | 0.99 |
| Discount for option Q-learning ($\gamma_Q$) | 0.9 |
| Buffer size | 500,000 |
| Value training iterations | 500,000 |
| VPS training iterations | 500,000 |
| Option-DQN training iterations | 500,000 |
| Batch size | 128 |
| Maximum episode length | 500 |
| Value network learning rate | $1 \times 10^{-3}$ |
| VPS network learning rate | $1 \times 10^{-4}$ |
| Option Q-network learning rate | $1 \times 10^{-4}$ |
| Value network loss function | Smooth L1 loss |
| VPS network loss function | Smooth L1 loss |
| Option Q-network loss function | MSE loss |

*Table 12.* Hyperparameter settings for VPS-based training in Atari 2600.

## C.2. Supplementary Experimental Results

**MiniGrid-KeyLock Random-Walk Success**

This result supplements the state-coverage and Q-learning results by measuring whether option-augmented random walks can complete the full key-door-goal sequence within a fixed horizon. The distribution shows that VPS options improve goal-reaching under random walk, while Eigenoption performance can drop for larger $k$ because higher-frequency Laplacian eigenvectors make options less bottleneck-focused.

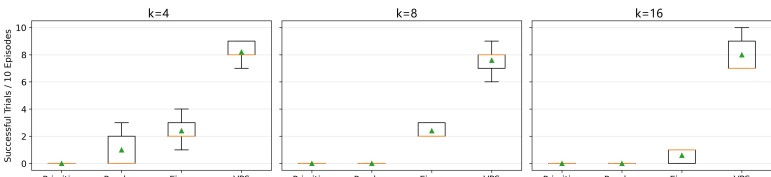

*Figure 14.* Successful-episode distribution in MiniGrid-KeyLock under option-based random walk with $k$ pairs of options. At each step, the agent uniformly selects among 6 primitive actions and all startable options. We evaluate 5 seeds with 10 episodes per seed (max 200 steps). Box: Q1–Q3; center line: median; triangle: mean; whiskers: 1.5×IQR.

**VPS Distribution in Other MiniGrid Environments**

These additional heatmaps show that VPS highlights bottleneck-like regions beyond MiniGrid-Rooms, including doorway and corridor structures in MiniGrid-FourRooms and Maze.

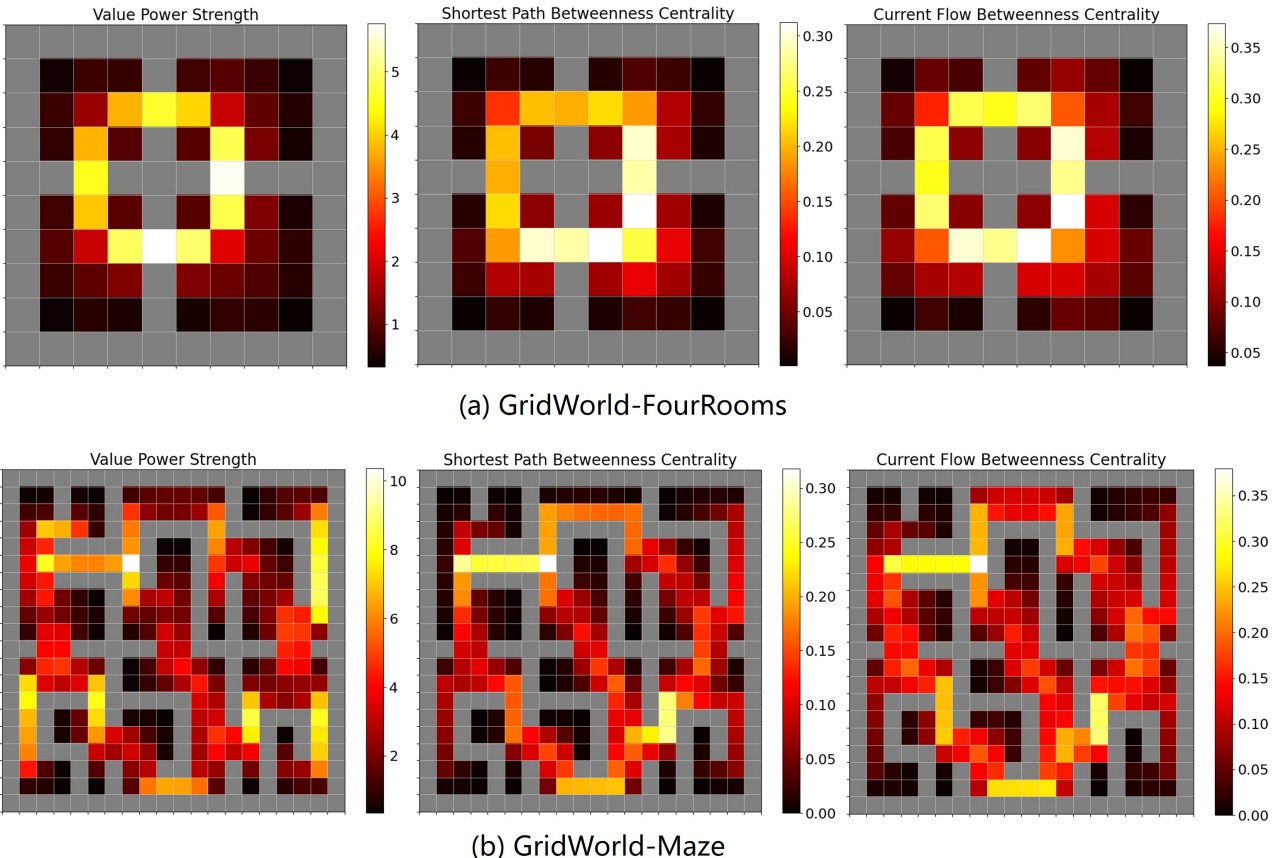

(a) GridWorld-FourRooms

(b) GridWorld-Maze

*Figure 15.* VPS Distribution in MiniGrid-FourRooms and Maze

## A Group of 20 VPS Options in MiniGrid-Rooms

This option set illustrates that different VPS-induced intrinsic rewards produce diverse, interpretable behaviors while remaining anchored to bottleneck-related regions.

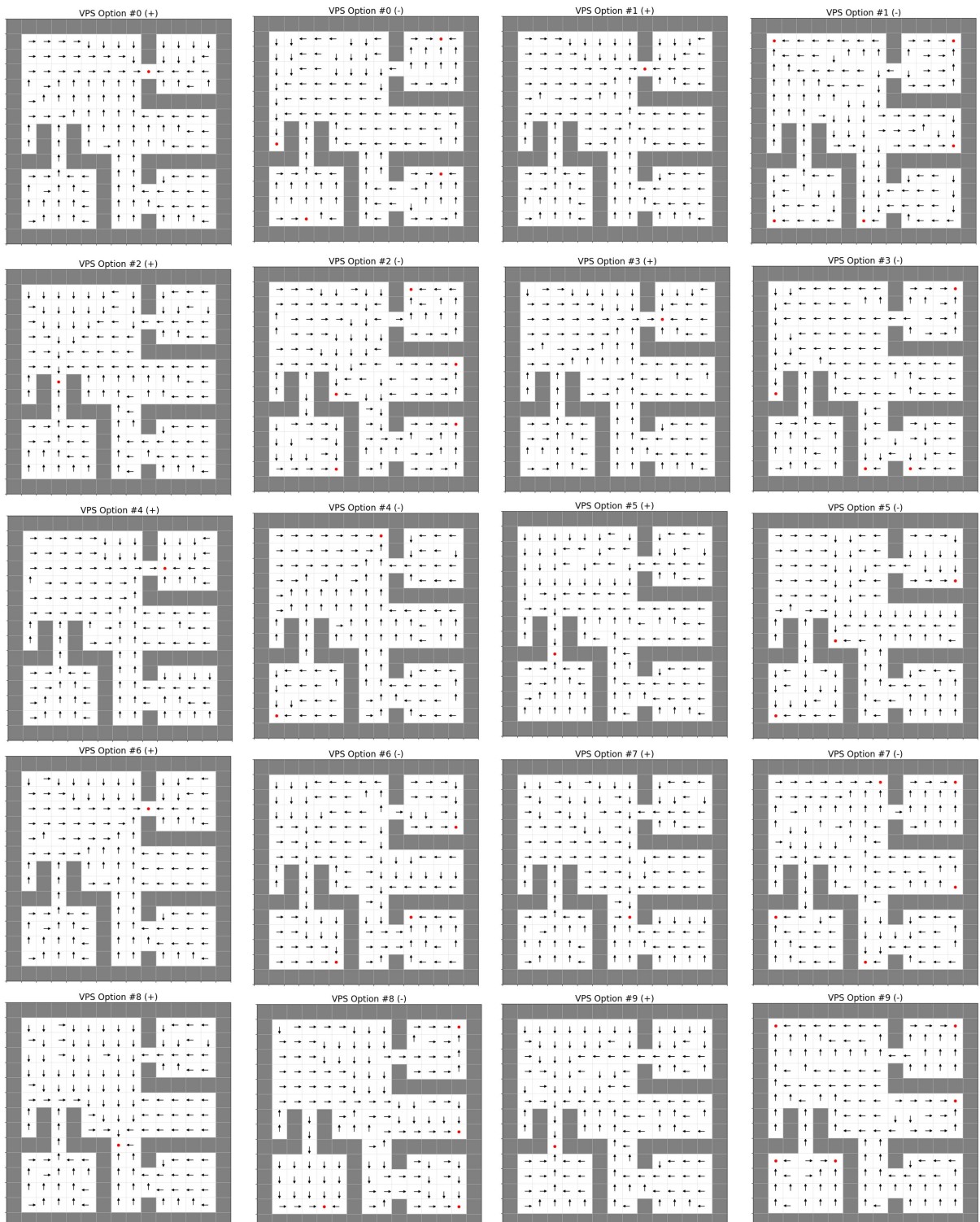

*Figure 16.* VPS Option Set #0 in MiniGrid-Rooms

## DIAYN Rollouts in PointMaze-FourRooms

These qualitative rollouts show reward-free DIAYN skills under the same maze layout used in the downstream PointMaze comparison. They visualize the baseline skill repertoire learned without task rewards, complementing the VPS option rollouts in Figure 7 and the downstream return comparison in Figure 8.

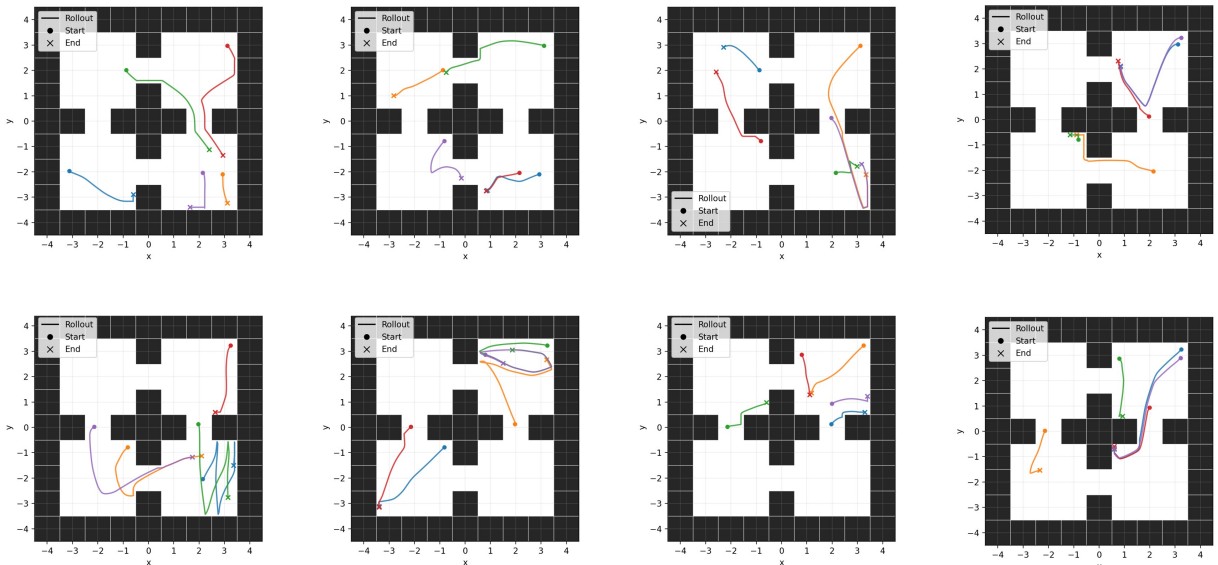

*Figure 17.* Representative DIAYN skill rollouts in PointMaze-FourRooms. Each panel visualizes trajectories produced by learned reward-free skills from different start states; dots and crosses mark rollout starts and endpoints.

## VPS Peaks in Atari-PrivateEye

In ALE/PrivateEye-v5, high-VPS frames appear around visually and behaviorally salient transitions, indicating states where reward diffusion is strongly concentrated.

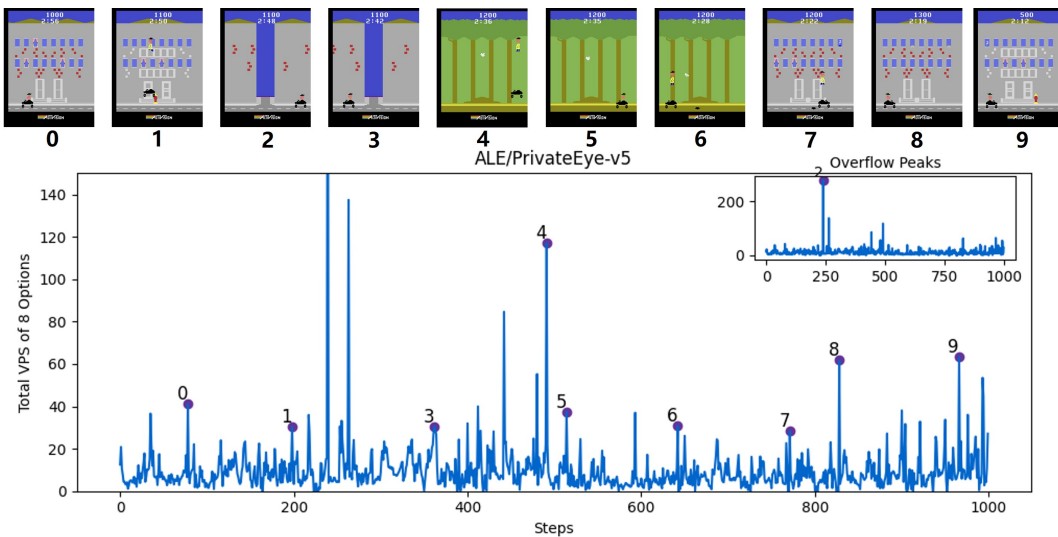

*Figure 18.* VPS Peaks in ALE/PrivateEye-v5

**VPS Peaks in Atari-MontezumaRevenge**

In ALE/MontezumaRevenge-v5, VPS peaks emphasize sparse, hard-to-reach visual states that are consistent with bottleneck-like exploration events.

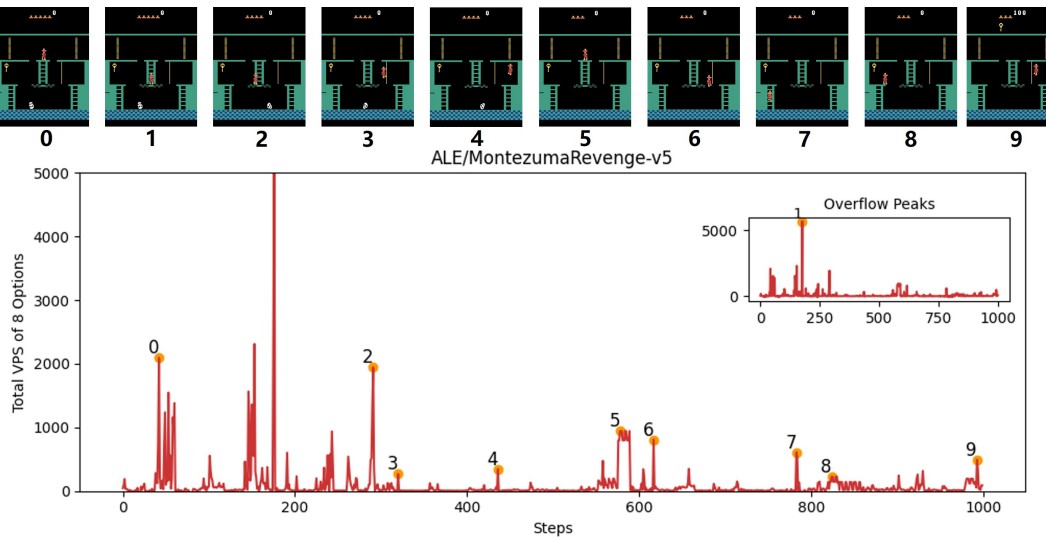

*Figure 19.* VPS Peaks in ALE/MontezumaRevenge-v5

**VPS Peaks in Atari-Freeway**

In ALE/Freeway-v5, high-VPS frames capture traffic-crossing states where small changes in position can strongly affect future reward propagation.

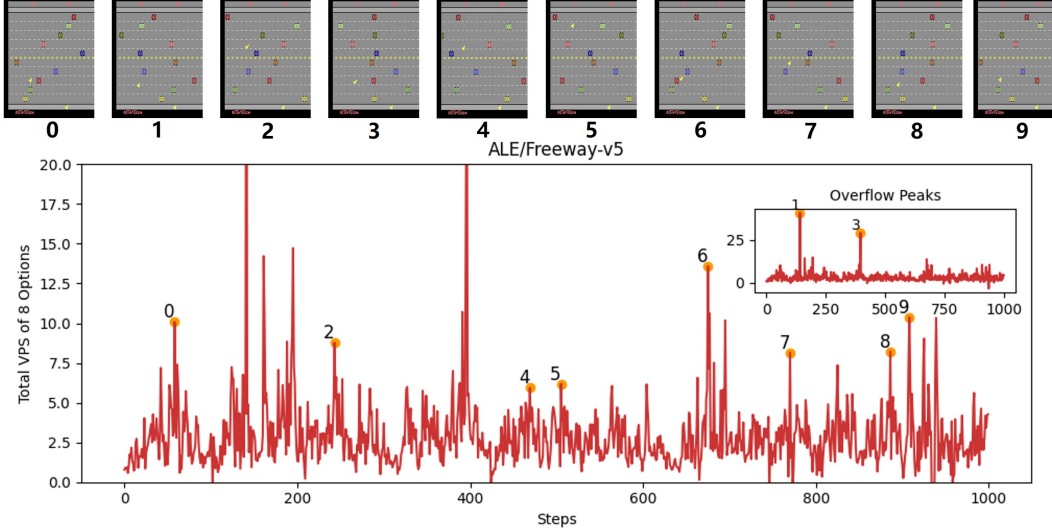

*Figure 20.* VPS Peaks in ALE/Freeway-v5

