# OpenReview forum: "Learning Interpretable Options by Identifying Reward Diffusion Bottlenecks in Reinforcement Learning"
_ICML.cc/2026/Conference — ICML 2026 regular_

### Official Review · Reviewer_4Paz · 2026-03-09

**Soundness:** 3
**Presentation:** 3
**Significance:** 3
**Originality:** 3
**Overall Recommendation:** 4
**Confidence:** 4

**Summary:**

The paper introduces Value Power Strength (VPS), a novel value-function-based metric designed to identify bottleneck states in Reinforcement Learning without explicit graph construction. The core contribution lies in a physical analogy: drawing a parallel between the Bellman equation and Kirchhoff’s current law to quantify "reward diffusion" through states. By learning auxiliary value functions from orthogonalized random rewards (via QR decomposition or RFF), the authors define VPS to capture the energy dissipation/flow concentration at bottlenecks. This metric is then leveraged to discover interpretable options that enhance exploration in both tabular and high-dimensional visual environments like Atari. The work effectively bridges the gap between traditional graph-theoretic bottleneck detection and modern deep RL function approximation.

**Compliance With Llm Reviewing Policy:**

Affirmed.

**Final Justification:**

After reading the other reviewers' comments and the authors' responses, I have decided to stay with my initial score.

**Key Questions For Authors:**

1. In highly irreversible MDPs (e.g., "falling to death" or one-way transitions), how does VPS maintain stability and accuracy in bottleneck identification?
2. Could the computational burden be mitigated by optimizing the weight-sharing mechanism between the main policy and auxiliary VPS heads?
3. How sensitive is the VPS metric to the variance ($\sigma^2$) of the random rewards?

**Limitations:**

The authors acknowledge the dependence on auxiliary tasks but do not provide a detailed analysis of the trade-off between the number of heads and exploration gain. Additionally, the theoretical constraints in directed/irreversible graphs remain a significant bottleneck for broader applications.

**Strengths And Weaknesses:**

Strengths:

1. The duality between Bellman residuals and circuit laws provides a principled and intuitive basis for bottleneck identification.
2. Unlike topological methods requiring full transition graphs, VPS integrates directly into the value-learning loop, enabling application to high-dimensional visual tasks (Atari).
3. The discovered options are anchored to semantically meaningful bottlenecks (e.g., doorways), offering better transparency than traditional black-box HRL methods.

Weaknesses:

1. Estimating VPS requires learning multiple auxiliary value heads from random rewards, which significantly increases the training FLOPs and memory footprint.
2. The convergence and spectral solutions rely on the assumption of reversible MDPs, which may not hold in many complex or directed environments.
3. The quality of bottleneck identification likely depends on the number of random rewards and the orthogonalization frequency, which is not fully explored.

---

> ### Author Rebuttal · Authors · 2026-03-26
>
> Dear Reviewer 4Paz,
>
> Thank you for your careful reading and constructive feedback. Below we respond point-by-point to your key questions and noted weaknesses.
>
> ## **1. Key Questions**
>
> ### **Q1. Irreversible MDPs**
>
> We thank the reviewer for raising this important point. Reversibility is indeed important for the spectral justification of QR rewards, but it is not required for the definition of VPS, the convergence of the online estimator, or the option-learning pipeline.
>
> **Spectral role only.**
> Reversibility appears only in Lemma 4.2 / Theorem 4.3, where we analyze QR rewards in a clean special case. By contrast, VPS is defined from the policy-induced one-step value discrepancy, and Proposition 1 does not require reversibility. Thus, when reversibility fails, what is lost is the clean spectral interpretation of QR rewards, not the validity or learnability of VPS itself.
>
> **Broader bottleneck semantics in non-reversible MDPs.**
> In reversible environments, high-VPS states often correspond to classical bottlenecks such as doorways. In non-reversible environments, they can also mark **directed interfaces** where discounted future reachability changes sharply, e.g., one-way doors, key pickup/consumption, or unlocking new regions. We also agree that some irreversible **failure boundaries** (e.g., falling to death or life loss) may be highlighted; in practice, this can be mitigated by masking these events as pseudo-terminals. This broader interpretation is consistent with our results: **MiniGrid-KeyLock** already contains irreversible key/lock transitions, and many **Atari** games contain one-way or irreversible transitions, yet VPS still highlights semantically meaningful states. We provide an additional MiniGrid-KeyLock VPS visualization here: [supplementary KeyLock VPS figures](https://anonymous.4open.science/r/supplementary-figures-for-response-1F32), and further Atari examples already appear in Appendix C.2 (Figures 17--19).
>
> **General interpretation.**
> For both reversible and irreversible MDPs, let $A := (I-\gamma P^\pi)^{-1} = \sum_{t \ge 0} \gamma^t (P^\pi)^t$. The $i$-th row of $A$ is the discounted future occupancy / reachability pattern from state $i$. For neighboring states $i,j$, the expected squared value difference under isotropic random probes is proportional to $\|A_{i,:}-A_{j,:}\|_2^2$. Thus, if a one-step transition sharply changes future reachability, VPS becomes large. In non-reversible MDPs, VPS therefore highlights transitions that induce large changes in long-horizon future structure.
>
> ### **Q2. Weight sharing and compute**
>
> Thank you for this helpful suggestion. We agree that stronger weight sharing is a promising way to reduce VPS overhead. Our current function-approximation formulation already uses a shared encoder with lightweight value/VPS heads rather than one full backbone per reward. Sharing more with the main policy could further reduce both FLOPs and memory, and we will clarify this in the revision.
>
> ### **Q3. Sensitivity to $\sigma^2$**
>
> This depends on the reward construction. For **tabular QR-based rewards**, $\sigma^2$ mainly scales the raw Gaussian matrix *before* orthogonalization, so after QR its effect on the final probe directions is limited. For **RFF-based rewards**, $\sigma^2$ controls feature bandwidth/frequency: larger $\sigma^2$ yields more local/high-frequency probes, while smaller $\sigma^2$ yields smoother/more global probes. At the same time, in our experiments we find that typical bottlenecks—such as room transitions in Atari or doorway states in MiniGrid—are robust to this choice: across a reasonably wide range of RFF variances, VPS still identifies these canonical bottlenecks.
>
> ## **2. Weaknesses**
>
> ### **W1. FLOPs and memory**
>
> Thank you for rasing this important point. With matched option count $K$:
>
> - **Random Option / Eigenoption**: roughly $K$ option heads
> - **VPS Option**: the same $K$ option heads **plus** one VPS/value-estimation module
>
> Thus, compared with Random Option / Eigenoption, the extra VPS cost is **additive rather than multiplicative in $K$**. We fully acknowledge this overhead. To reduce it, we are considering **low-rank / shared-basis value-function approaches** (thanks for your suggestion about weight-sharing).
>
> ### **W2. Reversibility assumption**
>
> Please see **Q1**.
>
> ### **W3. Probe count / orthogonalization frequency**
>
> Thank you for raising this practical point. In our experiments, QR-orthogonalized rewards are generated once before training and kept fixed, so there is no repeated re-orthogonalization. Figure 4(b) already shows how exploration efficiency changes with the number of VPS options. In additional results not shown in the paper, exploration efficiency typically increases with the number of VPS options and then saturates, with the saturation point determined by the number of salient bottlenecks in the environment. We will make this explicit in the revision.
>
> Thank you again and we wish you all the best.

---

> > ### Author Rebuttal · Reviewer_4Paz · 2026-04-01
> >
> > Thank you for the rebuttal. All of my concerns have been addressed.

---

> > > ### Author Response · Authors · 2026-04-01
> > >
> > > Dear Reviewer 4Paz,
> > >
> > > Thank you again for your time and helpful feedback. We are very encouraged by the positive assessment of the novelty of our work.
> > >
> > > At the same time, motivated by the reviewers' comments, we recognized that the downstream RL evaluation of VPS-option could be further strengthened. We therefore added new experiments comparing VPS-option, DIAYN [1], and a SAC baseline on sequential waypoint tasks in PointMaze-FourRooms:
> > >
> > > **https://anonymous.4open.science/r/point_maze_VPS_DIAYN-4C91**
> > >
> > > The preliminary results show that both VPS-option and DIAYN improve reward collection efficiency over the SAC baseline, while VPS-option performs better on the bottleneck-centric task. We believe this further supports the practical value of VPS-option beyond bottleneck visualization alone.
> > >
> > > If the paper is given an opportunity for revision, we will incorporate these additional results into the manuscript.
> > >
> > > Thank you again for your support and thoughtful comments. Wishing you all the best!
> > >
> > > ### Reference
> > >
> > > [1] Eysenbach, B., Gupta, A., Ibarz, J., & Levine, S. (2019). *Diversity is All You Need: Learning Skills Without a Reward Function*. ICLR.

---

### Official Review · Reviewer_3YCd · 2026-03-12

**Soundness:** 2
**Presentation:** 3
**Significance:** 2
**Originality:** 2
**Overall Recommendation:** 4
**Confidence:** 3

**Summary:**

The paper proposes Value Power Strength (VPS), a bottleneck metric defined as the expected squared value difference between a state and its successors. VPS is estimated by learning value functions from random rewards and is then used to construct options through potential-difference intrinsic rewards. Experiments on MiniGrid, Taxi, visual MiniGrid, and Atari show that VPS identifies doorway-like bottlenecks and can improve exploration in some settings.

**Compliance With Llm Reviewing Policy:**

Affirmed.

**Final Justification:**

I am raising my score from 3 to 4.

On W1, I reconsidered my initial stance: Integrating circuit-theoretic intuition, random-reward value probing, and potential-difference option construction into a coherent and principled bottleneck-discovery framework is a meaningful contribution, even if individual components have prior roots.

On W2, the additional Point Maze results and re-reading of the experiments convinced me that VPS captures richer structure.

On W3, the authors showed that VPS does not require reversibility, making the framework more general than I initially thought. This concern is resolved.

**Key Questions For Authors:**

1. Can you provide downstream control or exploration-performance comparisons in high-dimensional settings, rather than only bottleneck visualizations?
---
2. How sensitive is VPS to violations of the reversibility assumption in practice?

**Limitations:**

yes

**Strengths And Weaknesses:**

S1. The paper provides useful theoretical grounding through the convergence result for the online estimator and the spectral analysis linking VPS to Laplacian structure.

---

S2. The comparison with betweenness centrality baselines (Figure 2) helps contextualize VPS against established graph-based methods.

---

W1. VPS is closely connected to the local Dirichlet energy of a value function, as the paper itself notes in Eq. (10). The paper also builds on several existing lines of work, including spectral option discovery, random-reward value prediction, and potential-difference intrinsic rewards. As a result, the main contribution reads more as a new bottleneck-oriented interpretation and integration of related spectral/value-based ideas than as a fundamentally new option-discovery principle.

---

W2. The Atari experiments only visualize VPS peaks on human gameplay videos without measuring downstream RL performance. The tabular results are small-scale, and improvements over eigenoptions are marginal or inconsistent (e.g., Figure 6(c) shows no clear advantage in Taxi reward curves).

---

W3. The spectral analysis (Lemma 4.2, Theorem 4.3) requires reversible Markov chains, which rarely holds in practice. No empirical analysis is provided on how violations affect VPS quality.

---

> ### Author Rebuttal · Authors · 2026-03-27
>
> Dear Reviewer 3YCd,
>
> Thank you for your detailed feedback, and constructive criticism that helped improve the paper. Our responses are below.
>
> ## Key Questions
>
> ### Q1. Can you provide downstream control or exploration-performance comparisons in high-dimensional settings, rather than only bottleneck visualizations?
>
> We thank the reviewer for pointing out this important gap in the original submission. To partially address this concern, we added a new **Point Maze** experiment with both **continuous state space** and **continuous action space**, comparing **SAC** and **SAC + VPS option**. We also visualize the normalized RFF-reward-induced value functions together with the resulting VPS distribution. The supplementary results are available at: https://anonymous.4open.science/r/supplementary-experimental-results-751212314
>
> The results demonstrate that VPS-option can still identify bottlenecks and improve learning efficiency in a continuous-control setting. At the same time, we agree that this is still not a rigorous comparison against a broader set of more advanced baselines.
>
> The reason is that the main goal of this paper is to introduce the **RL-centric VPS concept** for bottleneck identification in MDPs, design the corresponding **VPS-option pipeline**, and provide an initial validation of its effectiveness. By contrast, final algorithmic performance depends not only on the option design itself, but also on the RL backbone, network implementation, and hyperparameters. Our current VPS-option implementation still has room for improvement in these aspects. In future work, we plan to refine the algorithmic implementation and compare it more systematically against stronger contemporary baselines.
>
> ### Q2. How sensitive is VPS to violations of the reversibility assumption in practice?
>
> Thank you for raising this important issue. The effectiveness of VPS is **not** impaired by non-reversibility. What changes is that, in non-reversible environments, the identified high-VPS states have broader semantics: rather than only corresponding to classical undirected bottlenecks, they can also capture more general **directed transition interfaces**. A more detailed discussion is provided in our response to Reviewer 4Paz, Q1 (Reviewer #4, at the bottom of the response page).
>
> ## Weaknesses
>
> ### W1. The contribution reads more as an interpretation / integration of existing ideas than as a fundamentally new option-discovery principle
>
> We appreciate the reviewer’s perspective, and apologize for possible confusion caused by our presentation. We would like to clarify that the core contribution is not simply an integration of existing option-discovery ingredients, but the introduction of a new RL-centric bottleneck quantity, VPS, inspired by circuit theory.
>
> The key is the analogy between node-wise power dissipation in electrical networks and reward diffusion in MDPs, through which we identify the RL counterpart of node power and define **Value Power Strength (VPS)** as a state-level bottleneck metric. Based on this quantity, we derive a bottleneck-oriented option design rule and provide an initial validation of the resulting VPS-option pipeline. To our knowledge, prior work has rarely approached bottleneck identification in MDPs through value-function-based reward-diffusion analysis in this way.
>
> This is different from the mainstream bottleneck-discovery literature, where bottleneck states are usually identified through **graph construction** and topological analysis. The spectral analysis, random rewards, and potential-difference intrinsic rewards mentioned in our paper are used to **justify, estimate, and operationalize VPS**; they do not define VPS itself. We agree, however, that the original submission could have stated this contribution boundary more sharply, and we will revise the paper accordingly.
>
> ### W2. High-dimensional experiments lack downstream RL evidence, and the tabular gains over eigenoptions are small or inconsistent
>
> This is indeed an important practical question. For the lack of downstream evidence in more complex settings, please see our response to **Q1** above.
>
> Regarding the modest gains over eigenoptions, we agree that stronger empirical improvements would have made the paper more convincing. The current paper mainly establishes VPS as a bottleneck-measure and provides an initial proof-of-concept implementation, rather than a fully optimized hierarchical RL algorithm. We will therefore revise the empirical claims more carefully: the current evidence supports the usefulness of VPS for bottleneck identification and for improving exploration in some settings, but does not yet establish broad superiority over strong spectral baselines across domains.
>
> ### W3. The spectral analysis requires reversibility, which rarely holds in practice, and no empirical analysis is provided on how violations affect VPS quality
>
> Thank you for your comments; please see our response to **Q2** above.
>
> Wish you all the best!

---

> > ### Author Rebuttal · Reviewer_3YCd · 2026-04-02
> >
> > I thank the authors for the thorough rebuttal and the additional Point Maze experiment. On W1, the circuit-theory reinterpretation is interesting, but my view that the novelty is primarily in the interpretation has not changed. On W2, I understand the current work is intended as a proof-of-concept, and after re-reading the experiments I think VPS captures richer structure than simple spatial bottlenecks, particularly in KeyLock and Taxi where object-state transitions are identified. On W3, I appreciate the clarification that reversibility is not required for VPS itself. However, since the QR reward design is motivated by the spectral result that does assume reversibility, I think a brief justification for why VPS remains effective without it would be valuable. I am maintaining my score.

---

> > > ### Author Response · Authors · 2026-04-05
> > >
> > > Dear Reviewer 3YCd,
> > >
> > > Thank you very much for your thoughtful follow-up. We especially appreciate your careful re-reading of the experiments and your observation that VPS captures richer structure than simple spatial bottlenecks, including object-state transitions in domains such as KeyLock and Taxi. We also appreciate your remaining concern on W3, which we agree deserves a clearer justification.
> > >
> > > Regarding W1, we fully understand your perspective that part of the contribution can be viewed as interpretive. We do not wish to overstate this point. What we would like to clarify is that VPS itself is not a standard quantity in RL, circuit theory, or graph theory, and, to our knowledge, has rarely been isolated and analyzed directly in these areas. In our view, the novelty lies in identifying, through the analogy among the three domains, a state-level quantity that reflects bottleneck strength and can be estimated through random-reward value probing. We agree that the downstream option design is more incremental, and there we indeed build on the potential-difference construction used in Eigenoption-style work [1]. We will revise the paper to delineate this boundary more carefully.
> > >
> > > On W3, we agree that our original rebuttal could have explained the non-reversible case more rigorously. Below we give a direct justification for why the QR reward design remains meaningful even when the MDP is not reversible.
> > >
> > > ### A non-reversible justification for QR reward probes
> > > Let $P$ be the policy-induced transition matrix of any finite MDP under a fixed policy, with discount $\gamma \in (0,1)$, and define $G = (I - \gamma P)^{-1}$. Since the spectral radius of $\gamma P$ is at most $\gamma < 1$, this inverse exists regardless of reversibility. For any reward vector $r$, the induced value function is $V = G r$. For a state $s$, define $L(s) = \sum_{s'} P(s'|s) (e_s - e_{s'})(e_s - e_{s'})^\top$. Then the VPS induced by reward $r$ can be written as $\phi_r(s) = r^\top G^\top L(s) G r$.
> > >
> > > We then obtain the following proposition.
> > >
> > > **Proposition.**
> > > Let $Q = [q_1, \ldots, q_k]$ be obtained by QR orthogonalization of an i.i.d. Gaussian matrix, so that the columns are orthonormal random reward probes as in Eq. (16) of the paper. For any finite MDP, without assuming reversibility,
> > >
> > > $E \left[ \frac{1}{k} \sum_i \phi_{q_i}(s) \right] = \frac{1}{n} \operatorname{tr}( G^\top L(s) G ).$
> > >
> > > Equivalently, if $g(s)$ denotes the row of $G$ corresponding to state $s$, then
> > >
> > > $E \left[ \frac{1}{k} \sum_i \phi_{q_i}(s) \right] = \frac{1}{n} \sum_{s'} P(s'|s) \| g(s) - g(s') \|_2^2.$
> > >
> > > **Proof (sketch).**
> > > For each probe $q_i$, we have $V = G q_i$, hence $\phi_{q_i}(s) = q_i^\top G^\top L(s) G q_i$. Averaging over $i$ gives
> > > $\frac{1}{k} \sum_i \phi_{q_i}(s) = \frac{1}{k} \operatorname{tr}( G^\top L(s) G Q Q^\top ).$
> > >
> > > Because $Q$ is obtained by QR from an isotropic Gaussian matrix, $E[Q Q^\top] = \frac{k}{n} I$. Therefore
> > > $E \left[ \frac{1}{k} \sum_i \phi_{q_i}(s) \right] = \frac{1}{n} \operatorname{tr}( G^\top L(s) G ).$
> > >
> > > Expanding $L(s)$ then gives
> > > $\operatorname{tr}( G^\top L(s) G ) = \sum_{s'} P(s'|s) \| g(s) - g(s') \|_2^2.$
> > >
> > > This proposition does not rely on reversibility. It shows that QR random rewards admit a general interpretation: they provide isotropic, low-redundancy probes of the local sensitivity of the discounted future-reachability kernel $G = (I - \gamma P)^{-1}$. Thus, in non-reversible MDPs, VPS still measures how sharply discounted future reachability changes across one-step transitions, so high-VPS states naturally broaden from classical undirected bottlenecks to more general directed transition interfaces.
> > >
> > > We also agree that this theoretical clarification is stronger when paired with evidence in a practically non-reversible setting. To this end, we additionally conducted a comparison between VPS-option and DIAYN on PointMaze-FourRooms sequential waypoint tasks:
> > >
> > > **https://anonymous.4open.science/r/point_maze_VPS_DIAYN-4C91**
> > >
> > > This environment has continuous state and continuous action spaces, with observations $(x, y, v_x, v_y)$, and is not naturally modeled as a reversible MDP. Empirically, VPS-option still identifies bottleneck regions and shows a downstream advantage on the bottleneck-centric sequential waypoint task, while DIAYN performs better on the room-center task. This is consistent with the proposition above: in the non-reversible case, VPS remains effective because it responds to sharp changes in discounted future reachability, which in PointMaze correspond to bottleneck-like transition interfaces between rooms.
> > >
> > > We sincerely thank you again for raising this point. Your comment helped us realize that the non-reversible justification should be stated more rigorously and explicitly in the manuscript, and we will add this in the revision.
> > >
> > > Wish you all the best!
> > > ### References
> > > [1] Machado, M. C., Bellemare, M. G., & Bowling, M. (2017). *A Laplacian Framework for Option Discovery in Reinforcement Learning*. ICML.

---

### Official Review · Reviewer_hzn5 · 2026-03-12

**Soundness:** 3
**Presentation:** 3
**Significance:** 3
**Originality:** 4
**Overall Recommendation:** 4
**Confidence:** 3

**Summary:**

The paper introduces Value Power Strength (VPS), a value function-based metric for identifying bottleneck states for option discovery in hierarchical reinforcement learning. The authors bridge circuit theory and RL, connecting the Bellman expectation equation and Kirchhoff’s current law. The authors demonstrate that VPS can be estimated online and can be used to design intrinsic rewards that steer agents toward or away from bottlenecks.

**Compliance With Llm Reviewing Policy:**

Affirmed.

**Final Justification:**

The paper proposes Value Power Strength (VPS), a circuit-theoretic perspective on bottleneck identification that bridges Bellman equations and Kirchhoff's law, enabling subgoal discovery in deep RL. Originality is excellent, soundness and presentation are good, and the contribution is meaningful for hierarchical RL. My main concerns were robustness in non-reversible MDPs, computational cost, the narrow set of baselines, and the limitation to discrete action spaces. The rebuttal addressed most of these constructively. In a follow-up, the authors also ran the comparison against DIAYN I requested; with this, the empirical substance of the paper is improved but not yet comprehensive enough to justify a stronger recommendation. I will therefore maintain my weak accept recommendation.

**Key Questions For Authors:**

1. How does the wallclock time / sample complexity of learning $k$ different random reward value functions compare to the downstream benefits gained by the VPS? A comparison of total computational budget against the baselines would be good.
2. Given that Theorem 4.3 relies on the MDP being reversible, how robust is the VPS framework in directed or non-reversible environments?
3. How sensitive is the performance to the tuning of the termination condition parameter $N$?
4. The experiments currently focus on discrete action spaces. How do you think VPS would behave in continuous control domains?

**Limitations:**

yes

**Strengths And Weaknesses:**

### Strengths

+ The formulation of VPS, as mapping node power dissipation to the expected squared temporal difference of a value potential, is an interesting and creative perspective on representation learning in RL.
+ Identifying bottlenecks without building an explicit transition graph seems like a significant contribution. The authors bring graph-theoretic subgoal discovery into high-dimensional, deep RL envs, which has been a major challenge.
+ The paper is well-written.


#### Weaknesses

- The spectral justification for using QR-orthogonalised random rewards rests on the assumption that the MDP is reversible. Although the authors acknowledge this, I feel that  complex / real-world environments are rarely perfectly reversible. Further discussion on how the VPS metric would degrade in non-reversible MDPs would be great.
- I think another issue is the computational cost of training multiple value functions ($k$) induced by random reward signals to estimate the VPS field (acknowledged by the authors)
- The baselines are somewhat narrow (primitive actions, random options, and eigenoptions), though I do understand why the authors used these. However, I'd still say that comparing VPS against more contemporary unsupervised skill discovery / HRL methods would be good to see where the method stands.
- The authors scale VPS to continuous state spaces using Random Fourier Features, but all evaluated environments still rely on discrete action spaces.

---

> ### Author Rebuttal · Authors · 2026-03-27
>
> Dear Reviewer hzn5,
>
> Thank you for your careful reading and helpful feedback. Below we respond point-by-point to your questions and concerns.
>
> ## Key Questions
>
> ### Q1. Wallclock time and sample complexity.
>
> This is indeed an important practical question. In **discrete** settings, a natural comparison is to eigenoptions [1]: VPS-option avoids **explicit graph construction and eigendecomposition**, and instead discovers bottleneck structure directly from probe-induced value discrepancies. In **continuous** settings, the computational burden of VPS-option is most comparable to that of deep Laplacian-based options [2]. Under a shared-encoder implementation, VPS-option requires $k$ value heads, $k$ VPS heads, and $k$ option heads, while deep Laplacian options require $k$ feature heads and $k$ option heads. Thus VPS carries one additional family of $k$ heads, so its extra cost is **additive**, and its wallclock impact can be much smaller when the shared encoder dominates.
>
> From the perspective of **sample complexity**, VPS-option is comparable to eigenoptions [1]: both need enough data to learn $k$ representations to provide intrinsic rewards and $k$ temporally extended behaviors. The main additional cost of VPS is the probe-learning stage used to discover bottleneck. The relevant tradeoff is therefore whether this upfront cost is offset by downstream gains in exploration. We agree that the paper would be stronger with an explicit wallclock-budget comparison, and will briefly clarify it in the revision.
>
> ### Q2. Robustness in non-reversible environments
>
> In non-reversible environments, what weakens is mainly the **clean spectral justification for the QR reward** in Theorem 4.3, not the VPS framework itself: the VPS definition, online estimator, and option-construction pipeline do not require reversibility. A more detailed discussion supporting the robustness in non-reversible environments is provided in our response to Reviewer 4Paz, Q1 (Reviewer #4, at the bottom of the response page).
>
> ### Q3. Sensitivity to the termination condition parameter \(N\)
>
> We added a supplementary experiment under the same random-walk setting as Fig. 4(b), varying the termination probability \(1/N\): https://anonymous.4open.science/r/state-coverage-curve-with-different-N-6AE7, showing that, under equal-probability random walk over primitive actions / options, both overly small and overly large N reduce exploration efficiency.
>
> Although \(N\) does matter, VPS-options remain consistently better than primitive-action random walk across a fairly broad range of \(N\), indicating robust gains in **exploration efficiency**. Since prior work [3] argues that the learning advantage of option frameworks mainly comes from such exploration gains, these results also suggest corresponding learning benefits over a broad range of \(N\).
>
> ### Q4. Behavior in continuous control domains
>
> We appreciate the reviewer for raising this important question.  To address it, we added a new **Point Maze** experiment with both **continuous state space** and **continuous action space**, comparing **SAC** and **SAC+VPS**, and visualizing the normalized RFF-reward-induced value functions and their corresponding VPS distribution. The supplementary results are available at: https://anonymous.4open.science/r/supplementary-experimental-results-751212314
>
> These results provide initial evidence that VPS-option can extend beyond discrete-action domains, still identify bottleneck regions, and improve learning efficiency in a continuous-control setting.
>
> ## Weaknesses
>
> For **W1 (reversibility)**, **W2 (computational cost)**, and **W4 (continuous-control scaling)**, please see our responses to **Q2**, **Q1**, and **Q4** above, respectively.
>
> ### W3. Narrow baselines
>
> We thank the reviewer for highlighting this important question. The current paper mainly introduces VPS and provides an initial validation of the VPS-option pipeline, so it does not yet fully position the method against more contemporary unsupervised skill-discovery / HRL approaches. Our implementation is still a proof-of-concept, and final performance depends not only on option design but also on the RL backbone and hyperparameters. Nevertheless, stronger comparisons are essential, and in future work we plan to improve the implementation and compare it with more contemporary baselines.
>
> Wish you all the best!
> ### References
>
> [1] Machado, M. C., Bellemare, M. G., & Bowling, M. (2017). *A Laplacian Framework for Option Discovery in Reinforcement Learning*. In *Proceedings of the 34th International Conference on Machine Learning* (PMLR 70, pp. 2295–2304).
>
> [2] Klissarov, M., & Machado, M. C. (2023). *Deep Laplacian-based Options for Temporally-Extended Exploration*. In *Proceedings of the 40th International Conference on Machine Learning* (PMLR 202, pp. 17198–17217).
>
> [3] Nachum O, Tang H, Lu X, et al. Why does hierarchy (sometimes) work so well in reinforcement learning? arXiv preprint arXiv:1909.10618, 2019.

---

> > ### Author Rebuttal · Reviewer_hzn5 · 2026-04-01
> >
> > Thank you for your response. My remaining concern is W3. Your rebuttal acknowledges this gap but defers to future work. I still think at least one comparison against a contemporary unsupervised skill discovery method (e.g., DIAYN) is important for contextualising the practical value of VPS-option. Could the authors comment on whether such a comparison is feasible within the revision period?

---

> > > ### Author Response · Authors · 2026-04-04
> > >
> > > Dear Reviewer hzn5,
> > >
> > > Thank you again for your careful reading, thoughtful follow-up, and encouraging assessment of our paper. We are especially grateful for your positive evaluation of its originality. Your comments reflect a thorough understanding of our method, and your interest in the VPS-option has been very encouraging.
> > >
> > > We also apologize for the late reply. Your key question was whether we could provide, within the revision period, a comparison against a contemporary unsupervised skill discovery method such as DIAYN. Because the rebuttal system allowed only one response, we chose to complete the experiment first and then report the actual results, rather than reply earlier with only a tentative plan.
> > >
> > > Following your suggestion, we added a comparison between VPS-option and DIAYN on continuous-state, continuous-action sequential waypoint tasks in PointMaze-FourRooms. This sequential-waypoint setting is also a standard evaluation protocol used in the DIAYN paper [1]. The supplementary results are available at:
> > >
> > > **https://anonymous.4open.science/r/point_maze_VPS_DIAYN-4C91**
> > >
> > > ### Setup
> > >
> > > Both VPS-options and DIAYN skills are pretrained with the same budget of $10^6$ environment steps. We use 5 seeds and learn 8 skills/options per seed for both methods; for VPS, these are $4 \times 2$ dual-form options. After pretraining, a Double DQN meta-controller is trained over the learned SAC skills/options, each executed for 200 steps unless interrupted by waypoint acquisition. We also train a primitive-action SAC baseline.
> > >
> > > We evaluate two sequential waypoint tasks. In both, the agent starts from $(2,2)$ and each episode has a horizon of 1000 steps. A reward of $+1$ is given only when waypoints are visited in order; otherwise it is 0. If a VPS-option or DIAYN skill reaches a waypoint during execution, it is terminated immediately and the meta-controller reselects a skill/option.
> > >
> > > - **Task 1 (bottleneck-centric):** $(4,2) \rightarrow (6,4) \rightarrow (4,6) \rightarrow (2,4)$
> > > - **Task 2 (room-center-centric):** $(6,2) \rightarrow (6,6) \rightarrow (2,6)$
> > >
> > > ### Key hyperparameters
> > >
> > > | Method | Main settings |
> > > |---|---|
> > > | **VPS-option** | input: $(x, y, v_x, v_y)$; value net: multi-head MLP; VPS net: probabilistic multi-head MLP over log-VPS targets; `value_gamma=0.999`; `value_lr=3e-4`; `vps_lr=1e-3`; `option_gamma=0.99`; `sac_option_lr=3e-4`; `rff_sigma=0.001`; VPS outputs normalized before constructing potential-difference intrinsic rewards |
> > > | **DIAYN** | input: $(x, y, v_x, v_y)$; `sac_skill_lr=3e-4`; `skill_gamma=0.99`; `discriminator_lr=3e-4`; `reward_scale=1.0` |
> > >
> > > ### Main findings
> > >
> > > This experiment provides the comparison you suggested.
> > >
> > > First, both VPS-options and DIAYN skills substantially improve downstream reward collection speed over the primitive-action SAC baseline, showing clear practical value for both learned temporally extended behaviors.
> > >
> > > Second, the two methods show different strengths consistent with their learned semantics. VPS-options perform better on **Task 1**, where success depends on efficient traversal through bottlenecks. DIAYN performs better on **Task 2**, where broader within-room coverage is more beneficial.
> > >
> > > The visualizations support the same interpretation. Figure 2 shows that VPS-options tend to drive the agent toward bottleneck states (positive VPS-options) or room-corner regions (negative VPS-options), yielding clearer semantic structure and interpretability. Figure 3 shows that DIAYN skills are distributed more uniformly across rooms, but are less semantically structured.
> > >
> > > Therefore, although VPS-option does not uniformly dominate DIAYN, this comparison clarifies its practical value: VPS-option is particularly effective for bottleneck-sensitive tasks, whereas DIAYN is more naturally aligned with broader state-space coverage. This indicates that VPS-option is a structure-aware rather than generic skill-discovery method. We also conjecture that, with sufficiently many skills/options, DIAYN may eventually surpass VPS-option in overall coverage-oriented performance, since salient bottlenecks are limited while diversity-based skills can keep refining state-space partitioning. At the same time, VPS-option may be less sensitive to option count, because a small number of bottleneck-oriented options can already capture much of the environment’s key transition structure.
> > >
> > > If the paper is given an opportunity for revision, we will incorporate this experiment and its analysis into the revised manuscript. Regardless of the final outcome, we sincerely thank you again for your attention and constructive suggestions, which have helped us strengthen both the empirical evaluation and the overall quality of the paper.
> > >
> > > Wish you all the best!
> > > ### Reference
> > >
> > > [1] Eysenbach, B., Gupta, A., Ibarz, J., & Levine, S. (2019). Diversity is All You Need: Learning Skills Without a Reward Function. *International Conference on Learning Representations (ICLR)*.

---

### Official Review · Reviewer_w4Sx · 2026-03-15

**Soundness:** 3
**Presentation:** 3
**Significance:** 2
**Originality:** 3
**Overall Recommendation:** 5
**Confidence:** 3

**Summary:**

+ Summary & Contributions
	- The authors propose a "reward diffusion" approach for automatically identifying bottleneck states in a reinforcement learning (RL) environment.
	- The proposal hinges on the notion of "value power strength" which quantifies how much a state impacts the "flow" of value under a particular choice of behavior policy and reward function. The authors employ a random walk as the behavior policy and, to mitigate overfitting to any one particular reward function, offer a statistical method for generating distinct random rewards based on QR decomposition of a randomly sampled Gaussian matrix.
	- Experiments corroborate how the proposed VPS measure is able to highlight bottleneck states within tabular environments.
	- The authors further propose a hierarchical RL agent that uses pre-computed options to maximize/minimize VPS for a collection of randomly sampled rewards. Empirical results confirm that this hierarchical agent facilitates effective exploration and yields performance either comparable to or slightly surpassing that of classic eigenoptions.

**Compliance With Llm Reviewing Policy:**

Affirmed.

**Final Justification:**

Given the authors' rebuttal experiments addressing both scalability concerns as well as the ability to compete with alternative HRL baselines, I have updated my score.

**Key Questions For Authors:**

I believe my questions are clear from the comments above.

**Limitations:**

Yes

**Strengths And Weaknesses:**

+ Quality
	- Strengths
		- The authors offer a reasonable proposal for identifying bottleneck states without appealing to some of the more traditional graph-based methods seen in the RL literature, which are often heavily limited to tabular MDPs.
		- The theoretical results seem to be in order.
	- Weaknesses
		* Major
			- N/A
		* Minor
			- I'm always concerned about reversibility as an assumption on MDPs and for all MDP policies, but at least the authors acknowledge it (L310-312).

+ Clarity
	- Strengths
		- The paper is well-structured and easy to follow.
	- Weaknesses
		* Major
			- N/A
		* Minor
			- I imagine Section 4.1 prior to Definition 4.1 could be a bit boring for RL researchers reading this paper. While the detail is nice to see as a piece of culture, it distracts from the RL focus of this work and, for this audience, the particular form of Equation 13 doesn't need such a lengthy justification. I suspect much of that text could be relegated to the Appendix.


+ Originality
	- Strengths
		- To the best of my knowledge, this particular approach for identifying bottleneck states and constructing options that achieve good state coverage is novel.
	- Weaknesses
		* Major
			- N/A
		* Minor
			- If one expands the Bellman equation for $V_\pi(s)$ in Equation 13, I'm pretty sure one can obtain a term equal to the expected next-state value variance, which is an established MDP complexity measure known as the environmental norm [2]. Perhaps there is some useful perspective/analysis to be done based on that fact which would further motivate VPS in a more RL-first/-centric manner. Moreover, by expressing VPS in terms of value functions (the first moment of the return random variable) and return variance (which requires the second moment of the return random variable), the authors might be able to leverage the fact that higher moments of the return (beyond the first moment = traditional value function) obey their own Bellman equations [4].

+ Significance
	- Strengths
		- The experiments confirm that the options computed via random QR rewards and VPS encapsulate knowledge of bottleneck states and that knowledge can be utilized in tandem with hierarchical RL to achieve an agent which either performs equal to or slightly better than eigenoptions. As far as I can tell, nothing about this particular formulation or agent relies on the environment being tabular.
	- Weaknesses
		* Major
			- While the components of this paper seem scalable, it's unfortunate that there isn't much demonstrated scalability in this paper itself. Since there are other approaches that obtain options based on random rewards [1], I wonder if the QR rewards might be swapped out to demonstrate the benefits of VPS in more complex environments than Minigrid and the Taxi domain. The authors provide visualizations of Atari domains but don't seem to include any learning curves showing that this structure captured by VPS translates into performance improvements, which is the key metric at the end of the day.
			- It is a little concerning that, for the two domains whose learning curves are reported, we either see performance comparable to that of eigenoptions or only slightly better. If we don't see results that confirm how much more easily VPS-based options scale to more complex problems, the next best thing we could hope for is that VPS options considerably outshine eigenoptions in terms of performance.
		* Minor
			- For the authors' mentioned challenge in the conclusion regarding learning many value functions simultaneously, they may wish to look into generalized value functions (GVFs) [3,5].


+ Final Remarks
	- Overall, this paper is decently strong, but lacking in the empirical evaluation to provide concrete evidence of how much more scalable and/or performant VPS is over baseline eigenoptions.


+ References
	1. Eysenbach, Benjamin, Abhishek Gupta, Julian Ibarz, and Sergey Levine. "Diversity is all you need: Learning skills without a reward function." arXiv preprint arXiv:1802.06070 (2018).
	2.  Maillard, Odalric-Ambrym, Timothy A. Mann, and Shie Mannor. "How hard is my MDP?" The distribution-norm to the rescue"." Advances in Neural Information Processing Systems 27 (2014).
	3. Schlegel, Matthew, Andrew Jacobsen, Zaheer Abbas, Andrew Patterson, Adam White, and Martha White. "General value function networks." Journal of Artificial Intelligence Research 70 (2021): 497-543.
	4. Sobel, Matthew J. "The variance of discounted Markov decision processes." Journal of Applied Probability 19, no. 4 (1982): 794-802.
	5. Sutton, Richard S., Joseph Modayil, Michael Delp, Thomas Degris, Patrick M. Pilarski, Adam White, and Doina Precup. "Horde: A scalable real-time architecture for learning knowledge from unsupervised sensorimotor interaction." In The 10th international conference on autonomous agents and multiagent systems-volume 2, pp. 761-768. 2011.

---

> ### Author Rebuttal · Authors · 2026-03-27
>
> Dear Reviewer w4Sx,
>
> Thank you very much for your time and comprehensive feedback. Below we respond point-to-point to the weaknesses you raised.
>
> ## Quality Weaknesses
>
> ### 1. Reversibility assumption
>
> We agree that reversibility is restrictive. However, it is only used for the clean spectral justification of the QR-random reward design, and is not required for the definition of VPS, the convergence of the online estimator, or the VPS-option learning pipeline itself. We will clarify this point more explicitly in the revision. A more detailed discussion is provided in our response to Reviewer 4Paz, Q1 (i.e. Reviewer #4, at the bottom of the response page).
>
> ## Clarity Weaknesses
>
> ### 1. Section 4.1 before Definition 4.1 is too long for an RL audience
>
> Thank you very much for your suggestion. Our intention was to provide intuition for why Eq. 13 takes its particular form, but we agree that this discussion is longer than necessary for an ICML/RL audience. In the revision, we will shorten the pre-Definition 4.1 discussion in the main text and move part of the motivation to the appendix.
>
> ## Originality Weaknesses
>
> ### 1. Relation between VPS, environmental norm, and higher moments of return
>
> Thank you for this insightful suggestion. We agree that this perspective is meaningful and helps position VPS in a more RL-centric way.
>
> In our formulation, VPS is computed as $\mathrm{VPS}^\pi(s)=\mathbb{E}[(V^\pi(s)-V^\pi(s'))^2]$, where $s' \sim P^\pi(\cdot \mid s)$. Using $\mathbb{E}[(X-a)^2]=\mathrm{Var}(X)+(\mathbb{E}[X]-a)^2$, this becomes $\mathrm{VPS}^\pi(s)=\mathrm{Var}[V^\pi(s')]+(V^\pi(s)-\mathbb{E}[V^\pi(s')])^2$, where again $s' \sim P^\pi(\cdot \mid s)$. This shows that VPS contains a **local next-state value-variance term**. More precisely, this term is the squared distribution-norm of $V^\pi$ under the policy-induced next-state distribution at state $s$, while the environmental norm of Maillard et al. is the corresponding global hardness measure obtained by taking the maximum over state-action pairs. We agree that making this connection explicit would better motivate VPS in an RL-first manner.
>
> Your comment also gave us a useful perspective on the formulation itself. Since VPS is currently computed in two stages—first estimating $V^\pi$, then forming one-step value discrepancies—it suggests a cleaner future direction: relating VPS to second-order statistics of future return and seeking a more direct bootstrapped VPS-like quantity. We thank you for this suggestion; in the revision, we will briefly clarify both the environmental-norm connection and this higher-moment perspective.
>
> ## Significance Weaknesses
>
> ### 1. Limited demonstrated scalability in the current paper
>
> We agree that this is an important weakness of the original submission. To address it, we added a continuous-state, continuous-action **Point Maze** experiment comparing **SAC** and **SAC + VPS option** , and visualized the normalized RFF-reward-induced value functions and VPS distributions. The results show that the VPS-option pipeline can extend to this continuous domain, identify bottlenecks, and improve learning efficiency. The supplementary results are available at: https://anonymous.4open.science/r/supplementary-experimental-results-751212314. This is still a preliminary scalability result, not yet a full benchmark against advanced HRL baselines. Broader comparisons in more challenging settings remain important future work.
>
> ### 2. VPS-options do not yet clearly outperform baselines
> We agree that the current performance gap over eigenoptions is not yet as strong as one would ideally like. As discussed in [1], the main benefit of HRL often comes from **improved exploration**. In our paper, VPS-options show stronger **state coverage** and **goal-directed exploration success** in environments with pronounced bottleneck structure (see Figs. 4(b), 5, and 6(b)), suggesting a meaningful exploration advantage.
>
> At the same time, final learning performance depends on the RL backbone and hyperparameters, not only on option quality. We therefore view the current implementation mainly as a **proof-of-concept** instantiation rather than a fully optimized hierarchical RL algorithm. In follow-up work, we plan to further optimize VPS-option and combine it with RL backbones beyond Q-learning / DQN.
>
> ### 3. Suggestion to consider GVFs for learning many value functions simultaneously
>
> Thank you for this helpful suggestion. We agree that **generalized value functions (GVFs)** are highly relevant here. From a GVF perspective, our probe-specific value functions share the same dynamics but differ in their cumulants (the random reward probes), suggesting a natural way to improve scalability via shared representations across probes. We will mention this connection in the revision.
>
> Wish you all the best!
>
> ### Reference
> [1] Nachum O, Tang H, Lu X, et al. Why does hierarchy (sometimes) work so well in reinforcement learning? arXiv preprint arXiv:1909.10618, 2019.

---

> > ### Author Rebuttal · Reviewer_w4Sx · 2026-04-04
> >
> > I appreciate the authors' effort to provide a rebuttal experiment that demonstrates scalability. I also agree that perhaps the benefits of identifying and capitalizing on bottleneck states are sufficiently well-established that one should naturally expect a corresponding HRL agent to eventually outperform standard eigenoptions with VPS-options.

---

> > > ### Author Response · Authors · 2026-04-04
> > >
> > > Dear Reviewer w4Sx,
> > >
> > > Thank you very much for your time, detailed comments, and valuable suggestions. We particularly appreciate your perspective on the empirical evaluation and the practical positioning of VPS-option.
> > >
> > > We agree with your earlier observation that the original submission was limited in downstream performance evaluation and comparisons against stronger baselines. Although our rebuttal had already added a continuous PointMaze result to demonstrate scalability, we felt that a direct comparison against a contemporary unsupervised skill discovery method would further strengthen the practical positioning of VPS-option.
> > >
> > > To this end, we additionally conducted a comparison between VPS-option and DIAYN on continuous-state, continuous-action sequential waypoint tasks in PointMaze-FourRooms, a standard evaluation setting also used in the DIAYN paper [1]. The detailed setup and hyperparameters are provided in our reply to Reviewer hzn5's rebuttal comment. Briefly, both VPS-options and DIAYN skills are pretrained with the same interaction budget of $10^6$ environment steps and then evaluated on the same downstream sequential waypoint tasks under the same meta-controller setting. The supplementary results are available at:
> > >
> > > **https://anonymous.4open.science/r/point_maze_VPS_DIAYN-4C91**
> > >
> > > The new results further clarify the practical value of VPS-option. Both VPS-options and DIAYN skills improve downstream reward collection speed over the primitive-action SAC baseline. At the same time, the two methods show distinct strengths consistent with their semantics: VPS-options perform better on bottleneck-centric tasks, whereas DIAYN performs better on tasks favoring broader within-room coverage. This comparison suggests that VPS-option is particularly effective in bottleneck-sensitive environments.
> > >
> > > If the paper is given an opportunity for revision, we will incorporate this experiment and its analysis into the revised manuscript.
> > >
> > > We sincerely appreciate your comments, which helped us identify this empirical gap more clearly and motivated us to strengthen the paper further.
> > >
> > > Wish you all the best!
> > > ### Reference
> > >
> > > [1] Eysenbach, B., Gupta, A., Ibarz, J., & Levine, S. (2019). Diversity is All You Need: Learning Skills Without a Reward Function. *International Conference on Learning Representations (ICLR)*.

---

### Decision · Program_Chairs · 2026-04-30

**Decision:**

Accept (regular)

**Comment:**

The paper introduces Value Power Strength (VPS), a novel metric inspired by circuit theory (specifically the analogy between the Bellman equation and Kirchhoff’s current law) to identify bottleneck states in Markov Decision Processes (MDPs). By quantifying "reward diffusion," VPS identifies these bottlenecks without relying on explicit transition graph constructions, allowing it to scale to continuous and high-dimensional spaces. The authors leverage VPS to design intrinsic rewards for discovering interpretable options in Hierarchical Reinforcement Learning (HRL).

The reviewing committee generally found the paper to be highly original, well-written, and technically solid. Reviewers appreciated the creative bridge between graph-theoretic bottleneck discovery, electrical circuit theory, and deep reinforcement learning.  All reviewers praised the theoretical grounding and the conceptual novelty of mapping node power dissipation to value function discrepancies. The ability to discover semantically meaningful subgoals (e.g., doorways or key/lock interactions) without topological graphs was highlighted as a significant contribution to the HRL sub-field.

**Initial Weaknesses:** The primary concerns raised during the initial review phase centered around three main areas:

1.  **Empirical Evaluation & Scalability:** The initial submission lacked evidence of scalability to continuous control domains and lacked comparisons to contemporary unsupervised skill discovery baselines (relying mostly on eigenoptions and random options).
2.  **Reversibility Assumption:** Reviewers expressed concern that the theoretical justification for the QR-orthogonalized random rewards relied heavily on the MDP being reversible, which is rarely true in complex or real-world environments.
3.  **Computational Overhead:** The requirement to learn multiple auxiliary value functions induced by random rewards raised concerns about the computational cost and sample complexity.

**Rebuttal and Discussion Phase:**
The authors provided a comprehensive and highly effective rebuttal that actively engaged with the reviewers' concerns:
* **Expanded Baselines & Continuous Control:** The authors conducted new experiments in a continuous-state, continuous-action PointMaze environment, comparing the downstream RL performance of VPS-options against both a SAC baseline and DIAYN. These results successfully demonstrated the scalability of the approach and highlighted VPS's specific advantage in bottleneck-sensitive environments. Reviewer w4Sx explicitly raised their score to an Accept based on these additions.
* **Clarification on Reversibility:** The authors convincingly clarified that the reversibility assumption is only required for the clean spectral justification of the QR rewards, not for the computation or online estimation of VPS itself. They mathematically and empirically demonstrated how VPS captures directed transition interfaces in non-reversible MDPs.
* **Computational Cost:** The authors clarified the additive (rather than multiplicative) computational overhead compared to baseline methods and discussed promising weight-sharing optimizations.

The committee agrees that this is a technically solid paper with a highly original contribution to option discovery in HRL. While some reviewers (hzn5, 3YCd, 4Paz) maintained a Weak Accept score—primarily noting that the empirical evaluation, while improved, is still more of a proof-of-concept than a fully comprehensive benchmark against all state-of-the-art methods—the consensus is clear that the method is theoretically sound and the conceptual contribution is significant. Reviewer 3YCd also explicitly raised their score, acknowledging the value of integrating these ideas into a coherent framework.

Given the strong originality, the rigorous theoretical grounding, and the authors' diligent integration of new baselines and continuous-control experiments during the rebuttal, the paper comfortably meets the bar for acceptance. Nonetheless, the authors are encouraged to ensure that the clarifications regarding the reversibility assumption, the mathematical justification for non-reversible MDPs, and the new continuous-control PointMaze experiments comparing VPS against SAC and DIAYN are fully incorporated into the main text or appendix of the final manuscript.